# Adults on pre-exposure prophylaxis (tenofovir-emtricitabine) have faster clearance of anti-HIV monoclonal antibody VRC01

Yunda Huang [1,2,13] ✉, Lily Zhang[1], Shelly Karuna [1], Philip Andrew[3], Michal Juraska [1], Joshua A. Weiner [4], Heather Angier[1], Evgenii Morgan[1], Yasmin Azzam[1], Edith Swann[5], Srilatha Edupuganti[6], Nyaradzo M. Mgodi[7], Margaret E. Ackerman [4], Deborah Donnell[1], Lucio Gama[8], Peter L. Anderson[9], Richard A. Koup[8], John Hural[1], Myron S. Cohen[10], Lawrence Corey [1,11], M. Juliana McElrath[1,2,11], Peter B. Gilbert[1,12] & Maria P. Lemos [1,13]

Broadly neutralizing monoclonal antibodies (mAbs) are being developed for HIV-1 prevention. Hence, these mAbs and licensed oral pre-exposure prophylaxis (PrEP) (tenofovir-emtricitabine) can be concomitantly administered in clinical trials. In 48 US participants (men and transgender persons who have sex with men) who received the HIV-1 mAb VRC01 and remained HIV-free in an antibody-mediated-prevention trial (ClinicalTrials.gov #NCT02716675), we conduct a post-hoc analysis and find that VRC01 clearance is 0.08 L/day faster ($p = 0.005$), and dose-normalized area-under-the-curve of VRC01 serum concentration over-time is 0.29 day/mL lower ($p < 0.001$) in PrEP users ($n = 24$) vs. non-PrEP users ($n = 24$). Consequently, PrEP users are predicted to have 14% lower VRC01 neutralization-mediated prevention efficacy against circulating HIV-1 strains. VRC01 clearance is positively associated ($r = 0.33$, $p = 0.03$) with levels of serum intestinal Fatty Acid Binding protein (I-FABP), a marker of epithelial intestinal permeability, which is elevated upon starting PrEP ($p = 0.04$) and after months of self-reported use ($p = 0.001$). These findings have implications for the evaluation of future HIV-1 mAbs and postulate a potential mechanism for mAb clearance in the context of PrEP.

Globally, 1.3 million people were newly diagnosed with HIV-1 in 2022. Although testing, treatment, pre-exposure prophylaxis (PrEP) and other risk reduction strategies have slowed the spread of HIV-1[1], an effective preventive vaccine does not yet exist and many people remain at-risk. Passive immunization with broadly neutralizing monoclonal antibodies (mAbs) provides a novel approach as an additional potential HIV-1 prevention modality, alone or in combination with other existing prevention modalities.

Multiple HIV-1 mAbs have been assessed in human clinical trials in recent years[2,3]. VRC01 was originally discovered in a person living with HIV-1 for more than 15 years who maintained viral control without use

of antiretroviral therapy (ART)[4–7]. VRC01 binds the HIV envelope site that interacts with the CD4 molecule on target cells, has the capacity to neutralize a broad range of HIV-1 strains in vitro, and has demonstrated protection in multiple non-human primate challenge studies[8–10]. Additionally, the first efficacy study of an HIV-1 mAb was conducted in 4623 participants to evaluate VRC01 in two antibody-mediated prevention (AMP) efficacy trials in 2016–2020 (HVTN 704/HPTN 085 and HVTN 703/HPTN 081, ClinicalTrials.gov #NCT02716675 and #NCT02568215). Participants were randomly assigned to receive intravenous infusions (IVs) of VRC01 every 8 weeks at a dose of either 10 or 30 mg/kg or placebo, for 10 infusions in total, with dose and

schedule determined based on previous early phase clinical studies[11,12]. Although neither AMP trial demonstrated overall efficacy in reducing new HIV-1 diagnoses, approximately 75% prevention efficacy of VRC01 was observed against HIV-1 viruses that were sensitive toVRC01[13]. Participants in the AMP trials were provided counseling and access to free oral PrEP [tenofovir disoproxil fumarate (TDF)-emtricitabine (FTC)].

Several clinical studies using longitudinal serum samples from VRC01 recipients assessed the pharmacokinetics (PK) of VRC01 among individuals without HIV-1[11,12,14–16]. Overall, two-compartment models have been employed to describe the distribution of antibodies like VRC01 from the central compartment (e.g., blood and well-perfused organ tissues) into the peripheral compartment (e.g., less-perfused tissues), and redistribution from periphery back to the central compartment, with elimination from the central compartment. For passively infused VRC01, an elimination half-life of about 15 days has been observed in healthy adults[11,12,14–16], with body weight influencing the clearance rate of the antibody[16]. However, these studies did not examine VRC01 PK in individuals with evidence of PrEP use.

In 2020, in the US alone, there were about 1.2 million persons with an indication for PrEP, of which 25% have received a prescription[17]. Several studies highlighted demographic and situational characteristics associated with PrEP uptake and adherence[18,19], suggesting potential differences between PrEP and non-PrEP users that may be associated with mAb PK. Although the likelihood of an interaction between these two classes of drugs may be limited due to the distinct pathways regulating their metabolism, there have been multiple reports of mAb-small molecule drug interactions in several disease areas as summarized in Ferri et al.[20]. In rheumatology, the clearance of adalimumab, an anti-tumor necrosis factor-alpha (TNF-α) mAb used to treat adults with rheumatoid arthritis, is reduced by methotrexate, presumably through reducing the patient's ability to make anti-drug antibodies (ADAs) against adalimumab[21]. In oncology, Canakinumab increases clearance of drugs metabolized by drug transporter CYP3A4[22]; In cardiology, statins increase clearance of Evolocumab, an anti-proprotein convertase subtilisin/kexin type 9 (PCSK9) mAb in adults with hyperlipidemia, likely via inducing additional PCSK9 expression[23]. Likewise, the statins, Ezetimibe and Fenofibrate reduce the AUC of Alirocumab, another anti-PCSK9 mAb, likely via induction of PCSK9 expression[24]. Therefore, although concomitant use of PrEP and VRC01 was only observed in a subset of AMP participants who voluntarily elected to take PrEP, it is important to fill in the knowledge gap and investigate whether VRC01 PK in oral PrEP (TDF-FTC) users differs from that in non-PrEP users, and if present, explore potential mechanisms of such interactions between PrEP and VRC01.

In this study, among AMP participants (cisgender men and transgender persons who have sex with men) in the US who received VRC01 infusions and remained HIV-free throughout the study duration of 18 months, we compare PK features of VRC01 between PrEP and non-PrEP users. Our results indicate that VRC01 clearance from systemic circulation is faster and serum concentration over time is lower among PrEP users compared to non-PrEP users. We further explore potential molecular and physiologic mechanisms of such findings. Our investigation suggests that these PK differences are correlated with increased epithelial intestinal damage and permeability in PrEP users, likely associated with either clinical or subclinical side effects of PrEP.

## Results
### Study population and baseline characteristics
In the AMP study of VRC01 (vs. placebo), PrEP uptake was generally not high (<25% person years)[13]. However, we were able to identify 24 oral PrEP users (of TDF-FTC), and 24 non-PrEP users by sampling from VRC01 recipients enrolled in the US (see details of the sampling in Methods)[25]. A PrEP user was defined by meeting the following criteria during 18 months since enrollment in AMP: (1) accessed the PrEP

referral program on at least one occasion per self-report, (2) self-reported intermittent or continuous PrEP use, and (3) has ≥3 TDF-detectable dried blood spot (DBS) samples out of those collected at approximately all 10 infusion visits. Particularly, DBS samples were used to measure red blood cell concentrations of the PrEP metabolites emtricitabine-triphosphate (FTC-TP) and tenofovir diphosphate (TFV-DP) representing a combination of recent and cumulative dosing of PrEP, respectively[26]. As expected from voluntary uptake, patterns of PrEP usage varied across PrEP users, with 17 initiating PrEP at various times after study enrollment and 7 self-reporting PrEP use prior to enrollment (Fig. 1, red lines). Among these PrEP users, at the last visit with confirmed PrEP usage, their DBS samples contained a median TFV-DP level of 1145 fmol/punch (range 540–2437), suggesting an average of at least 4 doses/week in the past 6–8 weeks[26]. A non-PrEP user was defined by having no self-reported use of PrEP and no TDF-detectable DBS samples out of those collected at approximately all 10 infusion visits throughout the study (Fig. 1, blue lines).

The baseline characteristics of these 48 participants, with half receiving VRC01 at the 10 mg/kg dose and half at the 30 mg/kg dose, are summarized by PrEP user status in Table 1. All participants except one were assigned male sex at birth; all had no evidence of new HIV-1 diagnosis as assessed by antibody tests every 4 weeks throughout the study period. About a quarter of participants identified as Black or African American (29% in PrEP users; 25% in non-PrEP users). The distributions of baseline vital signs, safety hematological characteristics, proinflammatory marker levels and intestinal permeability markers measured prior to the first VRC01 infusion (i.e., baseline) were all within normal range. The distributions were also mostly similar between PrEP and non-PrEP users, except levels of IL-10 and IFN-γ were slightly higher among some PrEP users (Table 1, Supplementary Table 1, Supplementary Fig. 1), with weak correlations amongst the markers (Supplementary Fig. 2). Collectively, these baseline characteristics do not appear to discriminate the non-PrEP users (n = 24) and PrEP users overall (n = 24), or between the non-PrEP users (n = 24) and the subset of PrEP users who started PrEP after enrollment (n = 17) (Supplementary Fig. 3), suggesting general comparability between the two groups in baseline characteristics.

### VRC01 serum concentrations over time
Serum concentrations of VRC01 after each of the ten 8-weekly VRC01 infusions were measured by ELISA approximately every 4 weeks in these participants, with higher levels at 4 weeks and lower at 8 weeks (i.e., trough) post each infusion, as expected of antibody decay over time (Fig. 2a). The majority of the 48 participants (94%) received all 10 VRC01 infusions (Supplementary Table 2). The observed concentrations at Day 61 (5 days after the second infusion), geometric means of the observed concentrations across all 4- and 8-week post-infusion visits, and the observed concentrations at Week 88 (16 weeks after the last infusion) are descriptively displayed in Fig. 2b, showing generally lower levels [hence lower area under the curve (AUC)] and a steeper decay among PrEP users vs. non-PrEP users. A generally steeper decay between the 4-week and 8-week post-infusion visits after each infusion was also observed (two-sided Wilcoxon signed rank test p = 0.006, Fig. 2c).

Population pharmacokinetics (popPK) models were applied to analyze VRC01 concentrations over-time from both PrEP and non-PrEP users. The two-compartment popPK model was parameterized in terms of clearance from the central compartment (CL, L/day), volume of the central compartment (Vc, L), inter-compartmental distribution clearance (Q, L/day) and volume of the peripheral compartment (Vp, L), as have been done previously[14–16]. Overall, as expected of most mAbs, VRC01 exhibited a PK profile characterized by a rapid distribution phase followed by a slower decay elimination phase. The base model (without adjusting for participant-level characteristics) captured the kinetics of concentrations well and all PK parameters were

estimated with good precision (% relative standard error ≤30%) (Supplementary Table 3). The PK parameter estimates were mostly consistent with those previously reported for VRC01 among healthy adults with no evidence of PrEP use. However, in this analysis combining both PrEP and non-PrEP users, the estimates for CL and Vp appeared slightly higher, suggesting that PK could be different between PrEP and non-PrEP users.

## Effect of PrEP use on VRC01 pharmacokinetics (PK)

We used the Targeted Maximum Likelihood Estimation (TMLE) approach to compare VRC01 PK features between PrEP and non-PrEP users. TMLE is an alternative to standard linear or nonlinear regression that can reduce confounding bias with improved robustness and efficiency[27–29], especially for comparisons between non-randomized groups. Five individual-level PK features: clearance, Vp, distribution half-life, elimination half-life, and steady-state dose-normalized area under the curve were estimated from the base popPK model. Other individual-level PK features including Q and Vc were not considered because in the model only an overall population-level parameter was estimated due to limited inter-individual variabilities observed in the current data. Distribution half-life, elimination half-life and area under the curve were considered in addition to clearance and Vp because they are functions of the four PK parameters in the model and are often of high interest to characterize the decay rates in the distribution and elimination phases of an mAb, as well as the overall extent of exposure to an mAb. The distributions of these individual-level PK feature estimates by PrEP user status are shown descriptively without formal comparisons in Supplementary Fig. 4. The following baseline characteristics that were shown to influence individual-level PK were adjusted for in the TMLE analysis: age, body weight, race, baseline behavioral risk score[16], and creatinine clearance (Supplementary Figs. 5 and 6). Levels of two inflammatory markers (IFN-γ and IL-10) were also adjusted for as they appeared to differ between the two groups (Supplementary Fig. 1). Comparisons of the PK features between all PrEP (n = 24) and non-PrEP (n = 24) users via TMLE are displayed in Fig. 3 and Table 2. Specifically, the mean clearance rate of VRC01 was about 15% greater in PrEP users than in non-PrEP users (p = 0.002, multiplicity-adjusted p = 0.005) with an estimated mean difference of 0.08 L/day (95% confidence interval (CI): 0.03, 0.13), and the mean area under the curve of VRC01 serum concentrations was about 14% lower in PrEP users

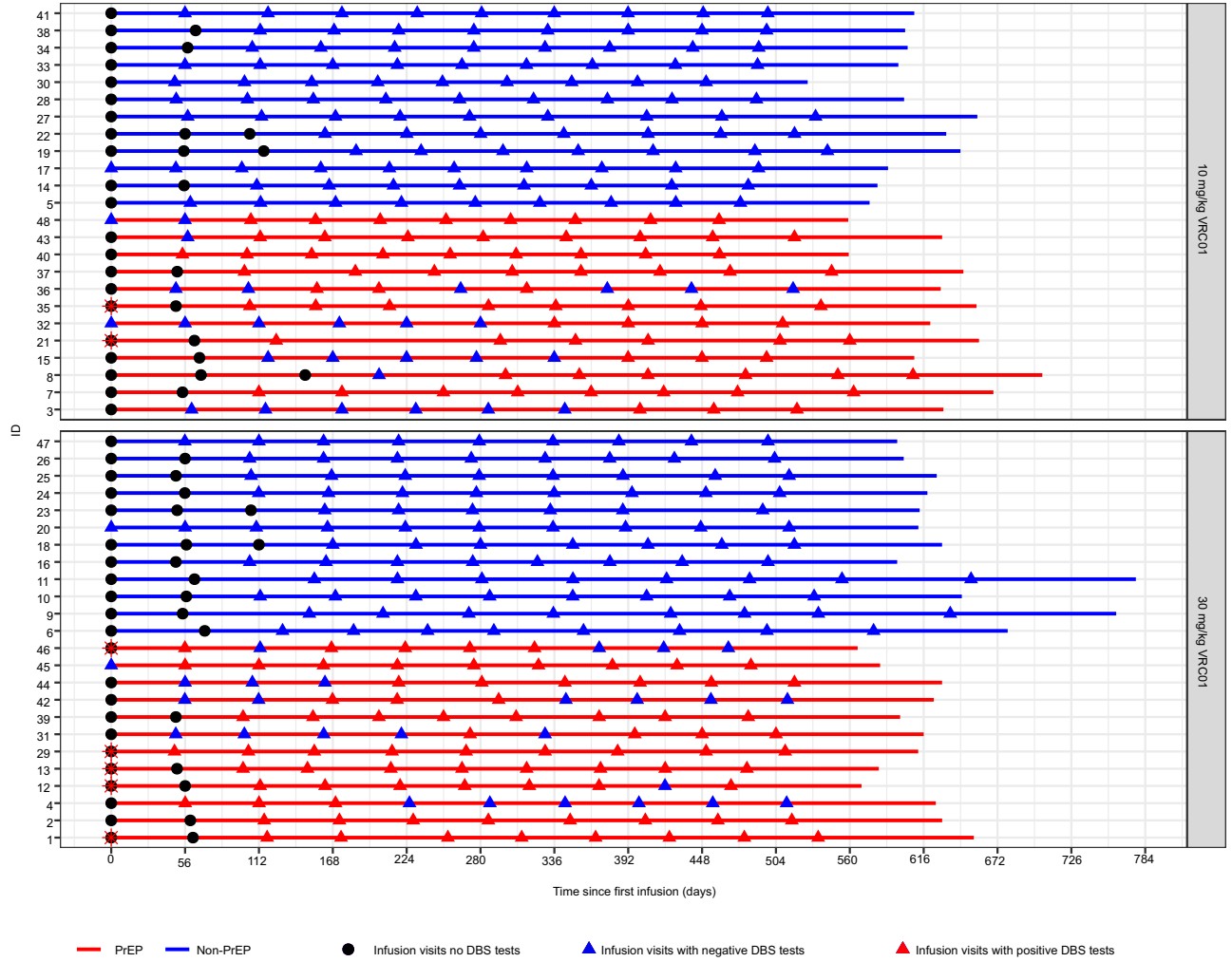

**Fig. 1 | Individual-level tenofovir disoproxil fumarate (TDF)-detection dried blood spot (DBS) testing data over time in PrEP (n = 24) and non-PrEP (n = 24) users by VRC01 dose group.** Timelines depict the follow up of study participants in both 10 mg/kg and 30 mg/kg VRC01 dose groups and their characterization as PrEP users (red lines) or non-PrEP users (blue lines). A participant was defined as a PrEP user if they met the following criteria during the study in HVTN 704/HPTN 085: (1) accessed the PrEP referral program on at least one occasion per self-report, (2) self-reported intermittent or continuous PrEP use, and (3) confirmation of ≥3 positive TDF-detectable DBS samples out of those collected at approximately all 10 infusion visits. A participant was defined as a non-PrEP user if they did not self-report any PrEP use and had no DBS samples tested positive for TDF. Circles depict infusion visits without DBS testing; red stars depict self-report of PrEP use prior to enrollment; blue triangles depict infusion visits with undetectable TDF; red triangles depict infusion visits with detectable TDF.

**Table 1 | Baseline characteristics of study participants by PrEP user status**

| Characteristics | Categories | Non-PrEP Users (N = 24[a]) | PrEP Users (N = 24[a]) |
|---|---|---|---|
| | | **N (%)** | |
| *VRC01 Dose (mg/kg) Received* | 10 mg/kg | 12 (50%) | 12 (50%) |
| | 30 mg/kg | 12 (50%) | 12 (50%) |
| *Demographics* | | | |
| Sex assigned at birth | Male | 23 (95.8%) | 24 (100%) |
| | Female | 1 (4.2%) | 0 (0%) |
| Race | Black | 6 (25.0%) | 7 (29.2%) |
| | Other | 18 (75%) | 17 (70.8%) |
| | | **Median (Range)** | |
| Body mass index (kg/m2) | | 25.8 (19.7, 37.6) | 25.4 (18.6, 34.5) |
| Weight (kg) | | 75.2 (59.9, 130.2) | 80.8 (55.7, 106.1) |
| Age (y) | | 31 (19, 50) | 27 (20, 43) |
| *Vital Signs* | | | |
| Diastolic blood pressure (mmHg) | | 74 (59, 89) | 76 (58, 94) |
| Systolic blood pressure (mmHg) | | 122 (109, 136) | 124 (95, 146) |
| Pulse rate (beats/min) | | 77 (51, 101) | 78 (50, 105) |
| Respiratory rate (breaths/min) | | 16 (12, 24) | 17 (12, 24) |
| Temperature (°C) | | 36.6 (36.0, 37.1) | 36.8 (35.8, 37.3) |
| *Safety hematology and chemistry labs* | | | |
| Alanine aminotransferase (units/L) | | 20 (8, 87) | 20 (7, 70) |
| Basophils (cells/mm$^3$) | | 31 (0, 140) | 23 (0, 100) |
| Creatinine clearance (CrCL) (mL/min) | | 126.2 (80.8, 212.3) | 128.9 (81.2, 180.9) |
| Eosinophils (cells/mm$^3$) | | 101 (38, 380) | 120 (10, 600) |
| Hematocrit (%) | | 43.6 (39.9, 48.0) | 44.1 (40.2, 48.5) |
| Hemoglobin (g/dl) | | 14.6 (12.8, 15.6) | 14.5 (12.9, 16.3) |
| Lymphocytes (cells/mm$^3$) | | 1932 (1043, 3058) | 1800 (920, 3759) |
| Erythrocyte mean corpuscular volume (fL) | | 90.4 (80.0, 100.3) | 91 (84.6, 96.7) |
| Monocytes (cells/mm$^3$) | | 455 (211, 740) | 485 (260, 3160) |
| Neutrophils (cells/mm$^3$) | | 3192 (1712, 5988) | 3370 (1138, 5483) |
| Platelets (10$^3$/mm$^3$) | | 229.5 (148.0, 335.0) | 227.5 (169.0, 432.0) |
| Leukocytes (10$^3$/mm$^3$) | | 5.9 (4.2, 9.8) | 6 (3.1, 9.9) |
| *Inflammatory markers[a] (pg/mL)* | | | |
| IFN-γ | | 3.1 (0.7, 11.6) | 5.5 (0.9, 454.9) |
| IL-6 | | 0.4 (0.1, 1.3) | 0.4 (0.1, 2.9) |
| IL-8 | | 13.6 (4.8, 91.3) | 10.4 (8.0, 53.2) |
| IL-10 | | 0.3 (0.1, 0.5) | 0.4 (0.1, 2.8) |
| TNF-α | | 1.2 (0.6, 13.2) | 1.3 (0.8, 14.8) |
| *Intestinal permeability[a]* | | | |
| Intestinal Fatty Acid Binding protein (I-FABP) (pg/mL) | | 710.1 (295.6, 3453.0) | 980.2 (230.3, 3169.3) |
| Lipopolysaccharide Binding Protein (LBP) (mcg/mL) | | 17.9 (4.8, 25.8) | 17.6 (9.2, 28.1) |
| *HIV-1 exposure* | | | |
| Behavioral risk score | | −0.9 (−2.2, 1.0) | −0.6 (−1.8, 0.8) |

A participant was defined as a PrEP user if they met the following criteria during the study in HVTN 704/HPTN 085: (1) accessed the PrEP referral program on at least one occasion per self-report, (2) self-reported intermittent or continuous PrEP use, and (3) confirmation of ≥3 positive Tenofovir Disoproxil Fumarate (TDF)-detection tests out of dry blood spot (DBS) samples collected at approximately all 10 infusion visits. A participant was defined as a non-PrEP user if they did not self-report any PrEP use and had no DBS samples tested positive for TDF.
[a]One non-PrEP user participant and one PrEP user participant who started PrEP after enrollment had missing baseline specimen and hence missing cytokine and permeability marker measurements.

than in non-PrEP users ($p < 0.001$, multiplicity-adjusted $p < 0.001$) with an estimated mean difference of 0.29 Day/mL (95% CI: 0.14, 0.45). Consistent results were observed in a sensitivity analysis that excluded data from one PrEP user (ID = 21) with unstable PK parameter estimates (Supplementary Table 4). Consistent results were also observed in a second sensitivity analysis that restricted the comparison between non-PrEP users ($n = 24$) and PrEP users who initiated PrEP after study enrollment ($n = 17$) (Supplementary Table 5). Lastly, since PrEP use could be intermittent for some of the PrEP users as shown in Fig. 1, additional analyses were performed to estimate the effect of current PrEP use on VRC01 PK that accounted for the time-varying status of PrEP use indicated by TDF-detectable DBS testing results (Supplementary Tables 6 and 7, Supplementary Figs. 7 and 8). The same trend was observed, showing a clearance rate that is 1.02 (95% CI: 1.02, 1.03) fold higher when PrEP was detected (vs. not detected).

**Translating PrEP effect on serum concentrations to PT$_{80}$, a biomarker of prevention efficacy**
As reported in Gilbert et al.[30], based on data from the AMP study, the predicted serum neutralization 80% inhibitory dilution titer (PT$_{80}$)

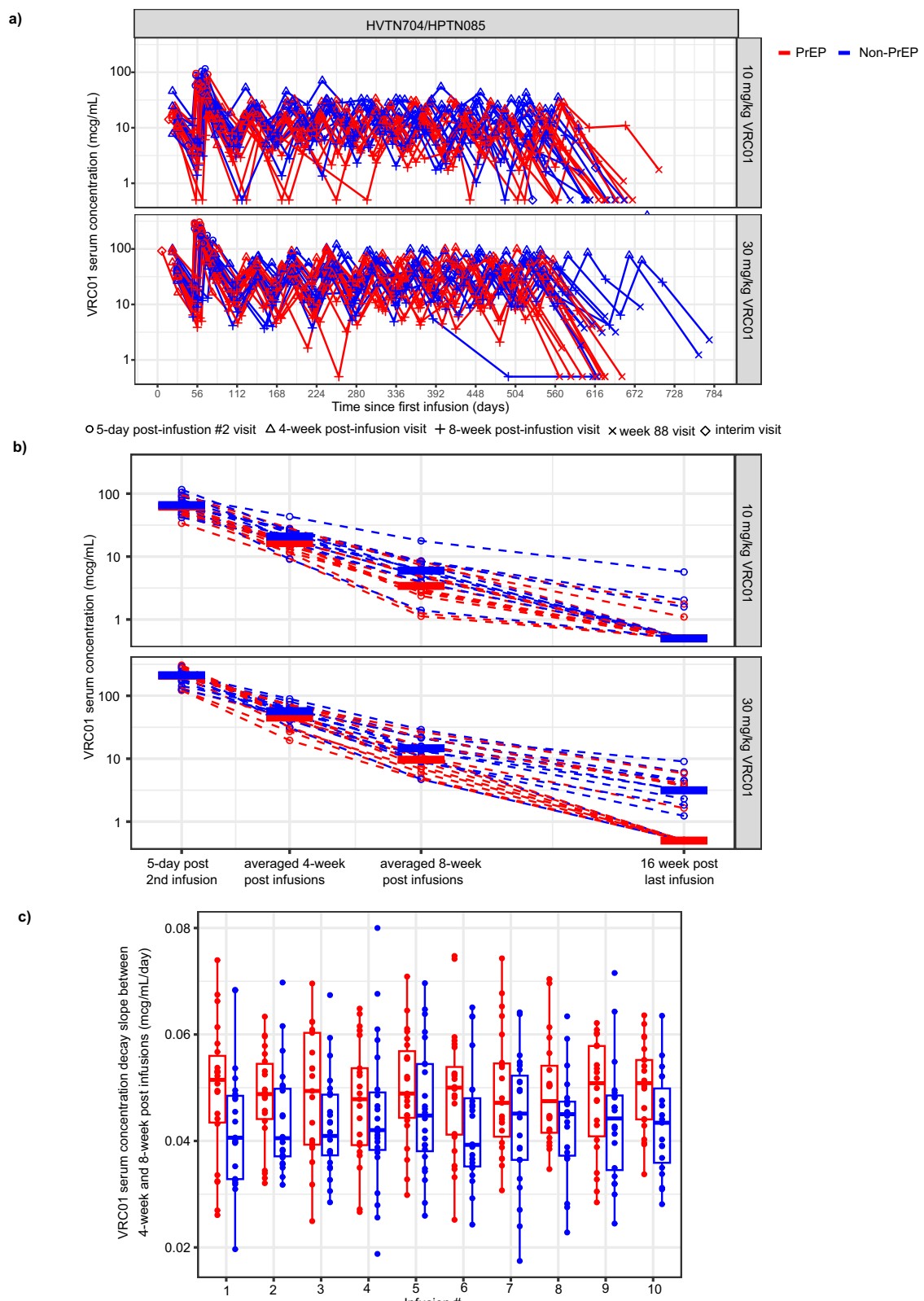

serves as a biomarker of HIV-1 prevention efficacy. $PT_{80}$ is typically calculated as the ratio between serum antibody concentration and the in vitro 80% neutralization inhibitory concentration ($IC_{80}$) of an antibody against a specific HIV-1 isolate. An average $PT_{80}$ of 200 against a population of likely exposing viruses was estimated to be required for 90% prevention efficacy against acquisition of these viruses.

We calculated $PT_{80}$ values to assess the impact of the reported PK differences on predicted VRC01-directed prevention efficacy between PrEP and non-PrEP users. Because ~75% prevention efficacy of VRC01 was observed against VRC01-sensitive viruses with $IC_{80} < 1.0$ mcg/mL in AMP[13], we examined $PT_{80}$ against such viruses based on the time-average VRC01 serum concentrations in steady state. Consistent with the differences in steady-state area under the curve, PrEP users had an

**Fig. 2 | Individual-level VRC01 serum concentrations and decay over time.**
**a** Observed VRC01 serum concentration (log$_{10}$-scale) over time in PrEP (*n* = 24), and non-PrEP (*n* = 24) users by VRC01 dose group. Serum samples were collected 4 weeks (open triangles) and 8 weeks after each infusion (crosses), as well as 5 days after the 2nd infusion at the Day 61 visit (open circles); last study visit with VRC01 serum concentrations measured was at the Week 88 visit (marked with x). All infusion visits were subject to a visit window between −7 and 49 days; all mid-points between infusion visits were subject to a visit window between −7 and 7 days; Day 61 visit was subject to a visit window between −1 and 3 days. **b** Distributions of VRC01 serum concentrations at Day 61, geometric mean concentrations across all attended 4-week and 8-week postinfusion visits, and concentrations at Week 88 in PrEP (*n* = 24) and non-PrEP (*n* = 24) users by VRC01 dose group. The horizontal bars

are the geometric mean of concentrations across PrEP (red) and non-PrEP (blue) users. **c** Distributions of decay slopes between 4-week and 8-week post-infusion visits in PrEP users (red) (*n* = 24) and non-PrEP (blue) users (*n* = 24) by infusion number. The decay slope after each infusion was calculated for each individual via dividing the difference in VRC01 serum concentrations (log$_{10}$-scale) between the 4-week and 8-week postinfusion visits by the actual number of days between the two visits. The mid-line of the box denotes the median and the ends of the box denote the 25th and 75th percentiles. The whiskers at the top and bottom of the box extend to the most extreme data points that are no more than 1.5 times the interquartile range (i.e., height of the box) or if no value meets this criterion, to the data extremes.

estimated 14% lower PT$_{80}$ values on average than non-PrEP users (*p* = 0.005) (Fig. 4a), assuming the same exposures to HIV-1 strains. For other viruses with more resistant neutralization IC$_{80}$, lower PT$_{80}$ values were observed, but the relative differences between PrEP and non-PrEP users were the same regardless of the neutralization sensitivity of the viruses (Fig. 4b, c). These differences highlight the potentially lower VRC01 PT$_{80}$-mediated prevention efficacy among PrEP users vs. non-PrEP users. However, it is important to note that the total likelihood of HIV-1 acquisition among these VRC01 recipients is still expected to be considerably lower among PrEP users (vs. non-PrEP users) because of their additional protection due to PrEP intake.

### Inflammatory markers and immune responses over time between PrEP and non-PrEP users

Inflammatory conditions can enhance the clearance of mAbs by affecting the activation of vascular endothelium[31], and modify receptor-mediated antibody clearance for antibodies such as anti-CD20 mAbs[32,33]. PrEP use could increase systemic inflammation, although conflicting results have been reported[34–38]. Therefore, in investigating potential mechanisms underlying the observed differences in VRC01 PK between PrEP and non-PrEP users, we first examined if proinflammatory markers IFN-γ, IL-6, IL-8, IL-10, and TNF-α were upregulated shortly after PrEP uptake (i.e., early visit on PrEP), and/or at the last study visit with a TDF-detectable DBS testing result (i.e., last visit on PrEP) (Fig. 5). The "early visit on PrEP" timepoint was defined as the earliest study visit after the first evidence of PrEP use based on self-report and DBS. For the 17 PrEP users who had no evidence of PrEP use prior to enrollment, this "early visit on PrEP" is ~4 weeks after the first evidence of PrEP use in AMP; for the 7 PrEP users who had evidence of PrEP use prior to enrollment, this "early visit on PrEP" is the enrollment visit in AMP. The "last visit on PrEP" is the last study visit with evidence of PrEP use, typically at the Week 72 visit for the 10th VRC01 infusion. The "last visit on PrEP' was a median of 438 days (interquartile-range (IQR): 233-546) since the self-reported date of PrEP initiation. We found that the levels of these cytokines were in normal range for most participants at these timepoints and we did not observe significant increases over time among PrEP users. Non-PrEP users also did not show elevations in these serum cytokines at the last study visit where DBS was assessed. These results indicated that the elevated clearance rate and lower area under the curve among PrEP users were likely not temporally associated with serum inflammatory markers.

Anti-Drug Antibodies (ADA) can also increase the clearance of infused mAbs, by enhancing IgG degradation via phagocytes[21,39]. However, as previously reported, ADA responses were not observed in VRC01 recipients in early phase 1 trials[11,12]. In AMP, ADA responses were observed in only 3% of 200 randomly sampled participants from both AMP trials; if present, ADA activity was transient and low, with titers less than 100[16]. Two of the 24 PrEP users and one of the 24 non-PrEP individuals in the current study had low observed ADA. Specifically, one PrEP user had detectable ADA at both the trough visit after the 6th VRC01 infusion and the last study visit, 32 weeks after the last infusion; another PrEP user had ADA responses only at the trough visit

after the 2nd VRC01 infusion. Lastly, the non-PrEP users had ADA at the last study visit. Together, these data suggest that clearance of ADA-VRC01 immune complexes was also an unlikely mechanism to explain the observed differential VRC01 clearance in PrEP vs non-PrEP users due to the minimal ADA observed among VRC01 recipients.

### Markers of hepatic and renal function over time among PrEP and non-PrEP users

Oral PrEP (TDF-FTC) is generally safe[40,41], but some people may experience side effects including gastrointestinal (GI) symptoms and in biomarkers (transaminases and creatinine) related to the liver and kidneys[42]. It is possible that PrEP-associated side effects may contribute to the observed differences in VRC01 PK between PrEP and non-PrEP users, since creatinine clearance could be associated with mAb clearance, and the liver appears to be a major site for catalysis of Fc-containing antibodies[43,44], such as VRC01.

We first examined whether there was any evidence of alterations to hepatic functions by assessing alanine transaminase (ALT) in serum based on available data. In PrEP users, we did not observe changes in ALT between baseline (*n* = 17, median=20 U/L IQR: [17,28]), early visit on PrEP (*n* = 13, 19 [19,22] U/L), or last visit on PrEP (*n* = 23, 22 [18.5, 26.5] U/L) (Fig. 6a). The ALT levels were also comparable to those of non-PrEP users at baseline (*n* = 24, 20 [14.8, 28.3] U/L), and at "last visit no PrEP" (*n* = 24, 21 [14.8, 28] U/L) (Fig. 6a). These data suggest no evidence of liver toxicity during the study period among the PrEP users receiving VRC01 included in this study. Therefore, the increased clearance of VRC01 in PrEP users was likely not associated with potential liver damage in people receiving both concomitant medications.

We next examined whether there was any evidence of kidney-related PrEP side effects by assessing creatinine clearance (CrCl) in serum over time. TDF is primarily eliminated in urine by a combination of glomerular filtration and active proximal tubular secretion[45], and can cause toxicity in proximal tubule epithelial cells of the kidney[46,47]. Serum creatinine can exhibit small increases in PrEP users within the first month after PrEP uptake, and long-term use can lead to sustained proteinuria in some individuals[48–51]. Therefore, the observed increase in VRC01 clearance and decrease in area under the curve among PrEP users could potentially be associated with decreased glomerular filtration rates (GFRs), which exclude antibodies from urine. To explore this renal mechanism, we assessed whether PrEP users in our cohort had evidence of reduced creatinine clearance or GFR at the early and late timepoints on PrEP. In the safety data available, we found no significant changes between CrCl levels measured at baseline (*n* = 17), at the "early visit on PrEP" (*n* = 13), or at the "last visit on PrEP" (*n* = 24) (Fig. 6b). Likewise, no significant changes in creatinine clearance values were observed among the 24 non-PrEP users at baseline and last visit. Additionally, no differences in GFRs using serum Cystatin C levels were observed at either timepoints in PrEP users and non-PrEP users, indicating no evidence of early kidney damage[52] (Supplementary Fig. 9). Together, these data suggest that the faster clearance of VRC01 and lower area under the curve among PrEP users were not temporally linked to potential kidney impairments at ~1 month or ~14 months of

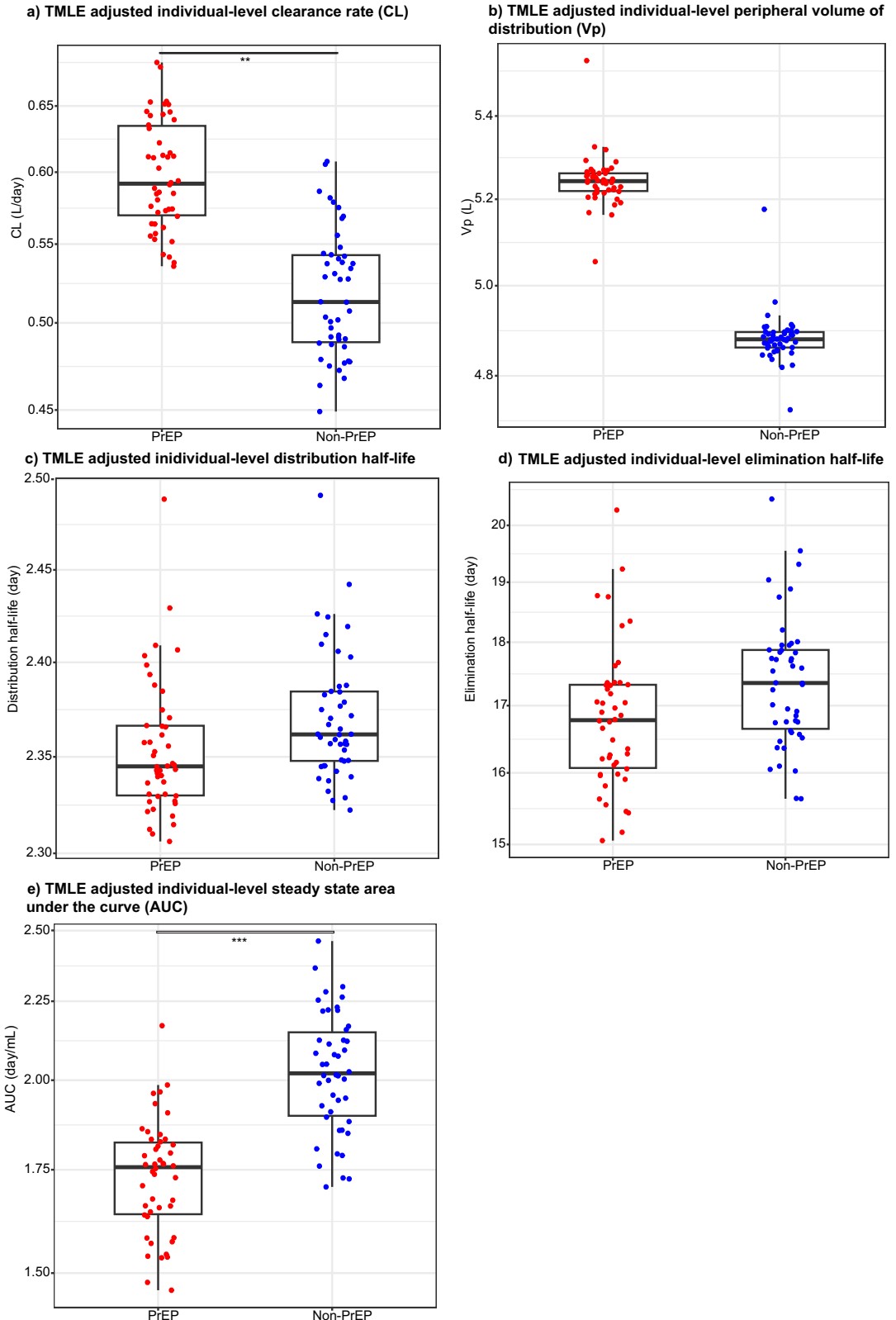

a) TMLE adjusted individual-level clearance rate (CL)

b) TMLE adjusted individual-level peripheral volume of distribution (Vp)

c) TMLE adjusted inidividual-level distribution half-life

d) TMLE adjusted individual-level elimination half-life

e) TMLE adjusted individual-level steady state area under the curve (AUC)

reported PrEP usage. We could not examine renal tubular secretion of mAbs and proteinuria, due to the absence of stored urine from the trial participants.

## Intestinal permeability over time between PrEP and non-PrEP users

GI side effects among oral PrEP users[53–55] include nausea, vomiting, diarrhea, stomach pain, and unintended weight loss. These are

commonly referred to as "PrEP Startup Syndrome," which occur more frequently within the first month after PrEP start, and tend to resolve on their own[46]. The exact mechanism causing these is unknown, but PK studies in animal and in vitro models have demonstrated that TDF-FTC accumulates in the intestine at higher concentrations than blood[56]. In addition, the clearance of VRC01, like endogenous IgG1, is believed to mostly occur through highly vascularized tissues such as the intestine[43,57,58]. Therefore, we explored the possibility that PrEP could

**Fig. 3 | Covariate-adjusted individual-level pharmacokinetic (PK) features of VRC01 serum concentrations among PrEP and non-PrEP users. a** Clearance (CL, L/day); **b** Volume of the peripheral compartment (Vp, L); **c** distribution half-life (day); **d** elimination half-life (day), and **e** steady-state area under the curve (AUC, day/mL). All PK feature estimates were adjusted for age, body weight, race (Black vs. other), creatinine clearance, behavioral risk score, IFN-g, and IL-10 via the targeted minimum loss-based estimation (TMLE) method as described in Methods with testing results presented in Table 2. The two-sided raw (adjusted) *p*-values for the comparisons were 0.002 (0.005) for CL (**a**), 0.44 (0.73) for Vp (**b**), 0.94 (0.94) for distribution half-life (**c**), 0.88 (0.94) for elimination half-life (**d**), and <0.001

(<0.001) for AUC (**e**). Each dot indicates the TMLE-adjusted PK feature estimate for each of the PrEP users (*n* = 24) and non-PrEP users (*n* = 24) as if all 48 individuals were PrEP users (red) or all were non-PrEP users (blue) under a causal framework. The mid-line of the box denotes the median and the ends of the box denote the 25th and 75th percentiles. The whiskers at the top and bottom of the box extend to the most extreme data points that are no more than 1.5 times the interquartile range (i.e., height of the box) or if no value meets this criterion, to the data extremes. *denotes 0.01 ≤ *p*-value < 0.05; ** denotes 0.001 ≤ *p*-value < 0.01; ***denotes *p*-value < 0.001.

**Table 2 | Comparisons of covariate-adjusted mean values of pharmacokinetic features between PrEP and non-PrEP users via the targeted minimum loss-based estimation (TMLE) method**

| PK feature | PrEP User Status | Mean (95% CI)[a] | Two-sided raw *p*-value[b] | Two-sided adjusted *p*-value[c] |
|---|---|---|---|---|
| Clearance [CL (L/day)] | PrEP | 0.60 (0.54, 0.66) | | |
| | Non-PrEP | 0.52 (0.46, 0.58) | | |
| | *Difference* | **0.08 (0.03, 0.13)** | **0.002** | **0.005** |
| Volume of the peripheral compartment [Vp (L)] | PrEP | 5.24 (2.59, 7.90) | | |
| | Non-PrEP | 4.88 (2.18, 7.58) | | |
| | *Difference* | 0.36 (−0.55, 1.28) | 0.44 | 0.73 |
| Distribution half-life (day) | PrEP | 2.35 (0.00, 8.10) | | |
| | Non-PrEP | 2.37 (0.00, 8.30) | | |
| | *Difference* | −0.02 (−0.42, 0.38) | 0.94 | 0.94 |
| Elimination half-life (day) | PrEP | 16.83 (4.40, 29.26) | | |
| | Non-PrEP | 17.39 (10.45, 24.34) | | |
| | *Difference* | −0.56 (−8.04, 6.91) | 0.88 | 0.94 |
| Steady-state area under the curve [AUC[d] (day/mL)] | PrEP | 1.74 (1.55, 1.92) | | |
| | Non-PrEP | 2.03 (1.83, 2.23) | | |
| | *Difference* | **−0.29 (−0.45, −0.14)** | **<0.001** | <0.001 |

All comparisons were adjusted for age, body weight, race, behavioral risk score, creatinine clearance (CrCl), IFN-γ, and IL-10 levels based on data from PrEP users (*n* = 24) and non-PrEP users (*n* = 24). One PrEP user and one non-PrEP user had missing baseline specimen hence missing IFN-γ and IL-10 levels. All TMLE estimation results of means were averaged over 20 runs with a fixed random seed on top of the 10-fold cross-validation estimation procedure to ensure stability of the estimates. A bootstrap procedure based on 500 datasets was used to calculate the empirical variances of the estimates for each group and to derive the 95% confidence interval, as well as to test for a non-zero mean difference between the two groups via the Wald test. The Holm method was used to adjust for multiple comparisons of the five PK features.
[a]Covariate-adjusted mean by targeted minimum loss-based estimation (TMLE) (See Methods for more details).
[b]Confidence intervals (CIs) and *p*-values based on empirical variances estimated via the bootstrap procedure. Bold = significant
[c]*P*-values adjusted by the Holm method to control for family-wise error rate.
[d]Area under the time-concentration curves divided by dose amount.

be associated with an increased clearance of VRC01 due to greater permeability of IgG-like antibodies into the intestinal lumen, where IgG can be degraded by intestinal proteases or be excreted in feces.

Due to its large size and complexity, the AMP study did not collect any mucosal samples to permit exploration of VRC01 clearance in intestinal secretions or intestinal pathology; however, we explored whether we could detect any evidence of increased intestinal permeability in blood. Under normal physiological conditions, intestinal Fatty Acid Binding protein (I-FABP or FABP2) is a 15kDa protein produced exclusively by intestinal epithelial cells and detectable in serum only at very low levels[59,60]. I-FABP does not contain a secretory signal sequence, so it is expected to enter systemic circulation following damage to the intestinal epithelium[60–64].

Among PrEP users, we observed a significant increase in I-FABP levels at both the "early visit on PrEP" (median: 1811 pg/mL, IQR: [1157, 2544] pg/mL, *p* = 0.04) and the "last visit on PrEP" (1585 [1307, 2299] pg/mL, *p* < 0.001) compared to the level at baseline (848 [561, 1512] pg/mL) (Fig. 6c). In addition, at the last visit on PrEP, the concentrations of TFV-DP among PrEP users correlated well with the I-FABP levels at that timepoint (*r* = 0.61 *p* = 0.004; Supplementary Fig. 10). In contrast, among the non-PrEP users, I-FABP levels maintained at similar levels over the study visits (baseline: 710 [535, 1165] pg/mL; last visit no PrEP: 718 [554, 1211] pg/mL). These results suggest that upon PrEP uptake and for months thereafter, PrEP users had evidence of increased

intestinal epithelial damage and increased permeability, whereas non-PrEP users did not.

In addition, Lipopolysaccharide (LPS) Binding Protein (LBP) is a 58kDa glycoprotein whose serum concentration increases when the mucosal barrier is damaged and bacterial LPS enters the bloodstream[65–70]. Since there was evidence of intestinal epithelial damage after PrEP initiation from the I-FABP results, we also examined LBP to determine the selectivity of intestinal barrier damage among PrEP users (Fig. 6d). Consistent with the lack of systemic proinflammatory cytokines (Fig. 5), we did not observe changes in LBP among PrEP users at baseline (20.8 [16.7, 24.0] mcg/mL), early visit on PrEP (14.9 [12.9, 18.3] mcg/mL), or after months of use (i.e., last visit on PrEP) (16.6 [13.1, 22.1] mcg/mL). The LBP levels were also comparable to those of non-PrEP users at the beginning of the trial (17.9 [13.7, 20.7] mcg/mL) and at last visit on PrEP (18.1 [11.9, 23.3] mcg/mL). These results suggest that although the intestinal epithelium could be chronically affected by PrEP use, as shown by increased serum I-FABP levels in PrEP users, we do not have evidence of LPS leakage into blood[71], as indicated by normal LBP.

### Impact of intestinal permeability on VRC01 pharmacokinetic (PK)

To solidify the hypothesis of intestinal permeability being associated with VRC01 clearance, we conducted a supportive analysis based on

**Fig. 4 | Individual-level predicted serum neutralization 80% inhibitory dilution titer (PT$_{80}$) biomarker among PrEP and non-PrEP users.** VRC01-specific PT$_{80}$ was calculated using time-averaged serum concentrations at steady state against HIV-1 viruses acquired among HVTN 704/HPTN 085 placebo recipients with neutralization sensitivity of **a** IC$_{80}$ < 1.0 mcg/mL; **b** 1.0 mcg/mL ≤ IC$_{80}$ < 3.0 mcg/mL; or **c** IC$_{80}$ ≥ 3.0 mcg/mL. The mid-line of the box denotes the median and the ends of the box denote the 25th and 75th percentiles. The whiskers at the top and bottom of the box extend to the most extreme data points that are no more than 1.5 times the interquartile range (i.e., height of the box) or if no value meets this criterion, to the data extremes.

modeling of VRC01 serum concentration data collected after the "early visit on PrEP". The reason to use data after, and not at or before this visit was because I-FAB-P levels were elevated at this timepoint only among PrEP users (not among non-PrEP users). To assess the impact of these elevated I-FABP levels on subsequent VRC01 serum concentrations, we excluded concentration data collected at or prior to the "early visit on PrEP" for PrEP users. In the same analysis, we also included VRC01 serum concentration data from non-PrEP users at both the "baseline no PrEP" and "last visit no PrEP" timepoints. We assessed the association between I-FABP levels measured at the "early visit on PrEP" with VRC01 PK assessed afterwards among both PrEP and non-PrEP users (Supplementary Fig. 8a–f). We found that I-FABP level at this timepoint was positively associated with the same participant's subsequent estimated CL rate of VRC01 (Fig. 7a), and negatively associated

with area under the curve of VRC01 (Fig. 7e), but not with the other PK features (Fig. 7b–d). These results further confirm the potential role of intestinal permeability in altering VRC01 PK.

## Discussion

To our knowledge, this is the first study investigating the effect and potential mechanisms of HIV-1 oral PrEP (TDF-FTC) on the pharmacokinetics of VRC01, an IgG1-backboned HIV-1 monoclonal antibody in healthy adults. Based on data from participants who received up to ten IV infusions of VRC01 during 80 weeks in the AMP trial, we found that VRC01 recipients who took PrEP exhibited significantly faster clearance rate of VRC01, compared to those who did not have evidence of PrEP use (mean 0.60 vs. 0.52 L/day, multiplicity-adjusted $p = 0.005$). In addition, PrEP users had a lower VRC01 exposure (i.e., dose-

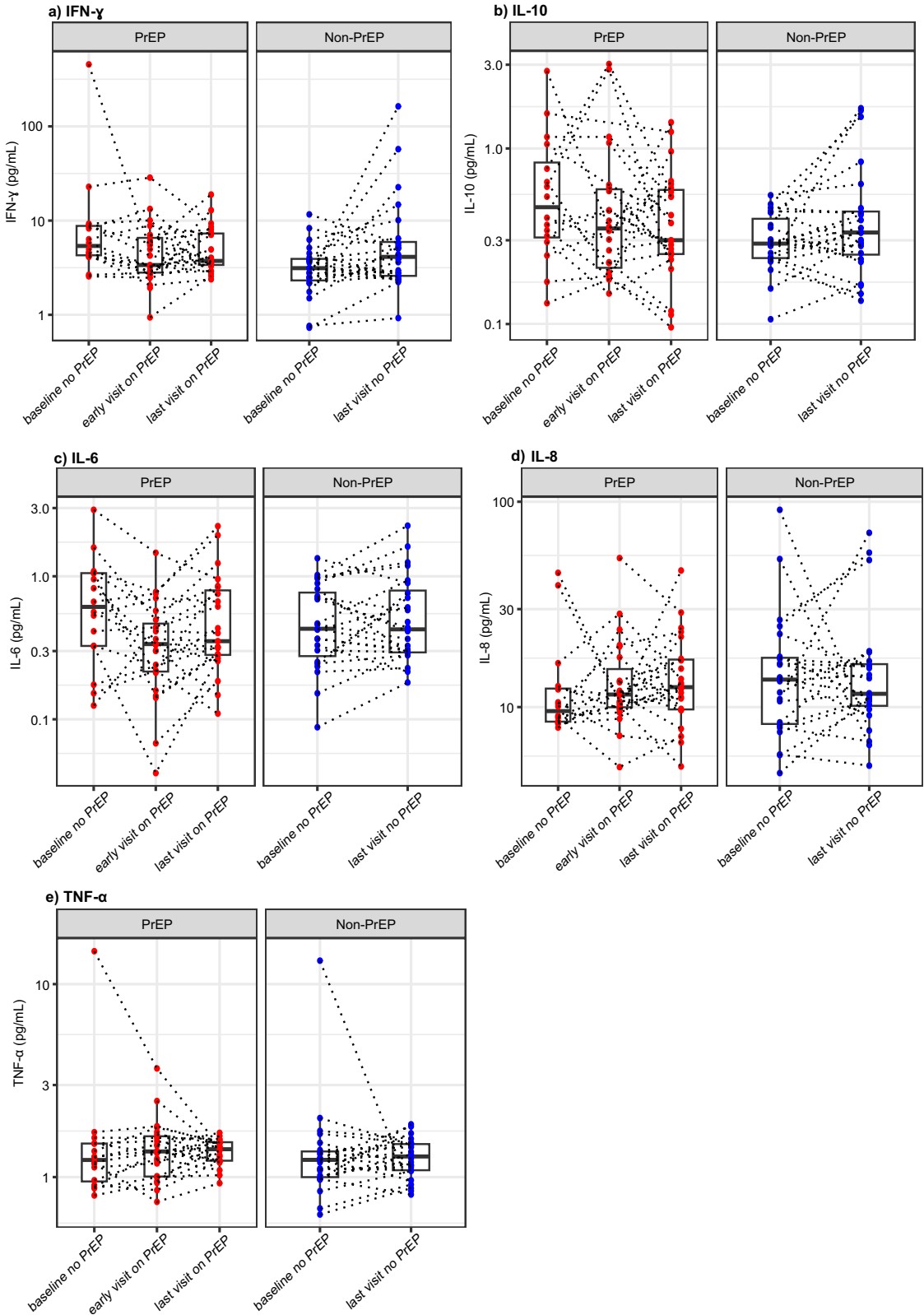

normalized area under the curve) compared to non-PrEP users (mean 1.74 vs. 2.03 day/mL, multiplicity-adjusted $p < 0.001$). These differences suggest faster elimination of VRC01 from the systemic circulation, and lower VRC01 serum concentrations over time in PrEP users vs. non-PrEP users.

Since different pathways are involved in regulating the metabolism, distribution and elimination of small molecule drugs and mAbs[72],

reports on potential PK interactions between drugs and mAbs have been limited and focused mainly on therapeutic (as opposed to prophylactic) mAbs. Three molecular mechanisms of drug-mAb interaction have been discussed in the literature:[20] (1) Immune-system-dependent drug-mAb interaction: where the immunosuppressive drug inhibits the formation of ADAs, and that slows the mAb degradation. (2) Cytokine-dependent drug-mAb interaction, where the metabolism

**Fig. 5 | Individual-level serum proinflammatory marker measurements among PrEP and non-PrEP Users. a** IFN-γ (pg/mL); **b** IL-10 (pg/mL); **c** IL-6 (pg/mL); **d** IL-8 (pg/mL); and **e** TNF-α (pg/mL) in serum were measured undiluted using a mesoscale multiplex proinflammatory panel 1 at multiple study timepoints. The sample sizes for each of the boxplots from left to right are $n = 16$ (PrEP users, baseline no PrEP), 23 (PrEP users, early visit on PrEP), 23 (PrEP users, last visit on PrEP), 23 (non-PrEP users, baseline no PrEP), and 24 (non-PrEP users, last visit no PrEP). The "early visit on PrEP" timepoint is the earliest study visit after the first evidence of PrEP use based on self-report and DBS. For the 17 PrEP users who had no evidence of PrEP use prior to enrollment, this "early visit on PrEP" is approximately 4 weeks after the first

evidence of PrEP use during the AMP study; for the 7 PrEP users who had evidence of PrEP use prior to enrollment, this "early visit on PrEP" is the enrollment visit. The "last visit on PrEP" visit is the last study visit with evidence of PrEP use, typically at the Week 72 visit for the 10th infusion, and a median of 439 days (IQR 233-546 days) since the self-reported date of PrEP uptake. The mid-line of the box denotes the median and the ends of the box denote the 25th and 75th percentiles. The whiskers at the top and bottom of the box extend to the most extreme data points that are no more than 1.5 times the interquartile range (i.e., height of the box) or if no value meets this criterion, to the data extremes.

of small molecule drugs is influenced by anti-IL-6, TNF-α or IL-1β mAbs that down-regulate associated drug transporters. And (3) Target-mediated drug-mAb interaction, where a small molecule drug increases the expression levels of the antigen that binds the mAb, and targets more of the mAb-antigen complex for degradation.

None of these three molecular mechanisms seemed likely to explain the PK interaction we observed between TDF-FTC and VRC01. Regarding the first mechanism, IV infusions of VRC01 rarely induced ADAs in healthy individuals[11,12,16]. Regarding the second mechanism, VRC01 did not modify cytokine concentrations, and we did not observe any evidence of proinflammatory cytokine changes in PrEP and non-PrEP users. In addition, VRC01, like most mAbs, has no known processing by the enzymes that activate and process TDF-FTC[45]. TDF-FTC is not known to impair P glycoprotein, an efflux transporter that could mediate drug interactions[73]. Lastly, regarding target-mediated drug interaction, direct VRC01 binding to sufficient HIV-1 to affect clearance is unlikely for our study participants who remained without HIV throughout the study period. Because unproductive HIV-1 exposures were possible in both PrEP and non-PrEP users, we adjusted for baseline behavioral risk score in the comparisons of VRC01 PK between PrEP and non-PrEP users, and the clearance and area under the curve differences remained significant after adjustment.

We hence focused our efforts on examining potential physiologic mechanisms, especially related to possible liver-, kidney- and intestine-associated side effects of PrEP usage. We did not find evidence of increased ALTs among PrEP users in our study cohort, suggesting that liver mAb clearance might be occurring normally. Kidney-associated side effects measured in terms of creatinine clearance correlated well with VRC01 clearance, but neither creatinine clearance nor Cystatin C GFRs had significant differences at -1 month or -14 months of reported PrEP usage, suggesting that the kidney glomeruli were unlikely sites for increased VRC01 clearance. Despite the known TDF-FTC toxicity in proximal tubule epithelial cells of the kidney[47], we were unable to explore directly any tubular secretion or reabsorption defects of the mAb, due to the lack of urine samples for assessment of proteinuria or antibodies in urine. Thus, we cannot completely exclude the possibility of differences in renal tubular secretion of VRC01 between PrEP and non-PrEP users.

In terms of intestine-associated side effects of PrEP use, we found a significantly increased level of intestinal epithelial damage as measured by I-FABP at -1 month and -14 months of reported PrEP use, compared to levels prior to PrEP uptake, indicating increased intestinal permeability at both timepoints among PrEP users. In addition, I-FABP levels were found to be positively correlated with the cumulative TFV-DP concentrations at the last visit on PrEP, suggesting a potential dose response. Interestingly, the magnitude of this I-FABP elevation is comparable to what was previously reported for people living with HIV on antiretroviral therapy[74]. In our study, such elevation was not observed among non-PrEP users during the same time period. These observations are consistent with previous reports of GI side effects for PrEP users, such as diarrhea, nausea, and unexplained weight loss[53–55], but appear to extend past the "PrEP Uptake Syndrome" phase. Additionally, we found that individuals with higher I-FABP levels had a subsequent increased clearance and decreased area under the curve of

VRC01. Putting all evidence together, we hypothesize that the mechanism of PrEP association with VRC01 pharmacokinetics may take place in the intestinal epithelium, where I-FABP is expressed.

As changes in serum I-FABP are associated with intestinal damage occurring in coeliac disease[60], inflammatory bowel disease[61], ischemia[62,63], and sepsis[64], we also explored the upregulation of LBP in serum. LBP increases after early sepsis[69], acute bacterial and viral gastroenteritis[69], HIV-1[75], and correlates with markers of paracellular intestinal transport[65]. Consistent with the lack of other proinflammatory markers, LBP was not up regulated after either -1 month or -14 months of reported PrEP usage, suggesting that pathways mediating intestinal bacterial LPS permeability, were not affected by PrEP. This result highlights that the intestinal permeability changes observed among PrEP users may be selective – TDF-FTC may alter specific transport or catabolic functions of intestinal epithelial cells but may preserve intact paracellular barriers[65] and endocytosis of lipid rafts[70,71] that have been shown to regulate intestinal LPS entry.

PrEP could affect several mechanisms of VRC01 transport in intestinal epithelium that require further study. On one hand, PrEP could be affecting the dynamic turnover of intestinal epithelial cells[76,77], shortening the villi, and reducing the absorptive surface area, where antibody is recycled from the lumen. This kind of absorptive effect has been demonstrated in mouse models for nelfinavir (NFV), indinavir (IDV), didanosine (DDI) and zidovudine (AZT)[78], and could have implications for the absorption of nutritional contents and other medications taken orally. On the other hand, PrEP could be modifying the endosomal transport and recycling of antibodies, primarily mediated by the neonatal Fc receptor (FcRn), which rescues albumin and up to 2/3 of the endogenously produced antibody from degradation[79,80]. It will be important to understand whether PrEP affects epithelial turnover and/or this recycling pathway, as new HIV-1 prophylactic mAbs in the pipeline use modifications to enhance FcRn recycling;[8] therefore, the clearance and half-life of the new modifications may also be differentially modified by PrEP. Studies of urine, intestinal biopsies, intestinal secretions, and blood from PrEP users and non-PrEP users might help elucidate whether these or other mechanisms are at play; and whether the VRC01 observation extends to other antibodies and to albumin. Additional research is also needed to understand whether our reported interaction between TDF-FTC and VRC01 extends to the long-acting injectable PrEP such as cabotegravir (CAB-LA) and lenacapvir[81,82].

This work has several limitations. First, because oral PrEP uptake was generally low in non-US sites of the AMP trials[13,25], we only included participants enrolled in the US. Therefore, our results may not be generalizable to other study populations with different intestinal microbiome and dietary habits. Second, our study cohort is relatively small and may suffer from low power to detect smaller differences potentially in CrCl, Cystatin C GFR, or ALT. Third, this was a post-hoc analysis to evaluate the association of PrEP use with VRC01 PK. We pursued interpreting the association as a causal effect via advanced statistical methods through confounder adjustment; however, unmeasured confounding could still have resulted in bias. Fourth, we did not investigate all potential mechanisms that catabolize antibody. Nonetheless, we provide the first indication that oral HIV-1 PrEP has an

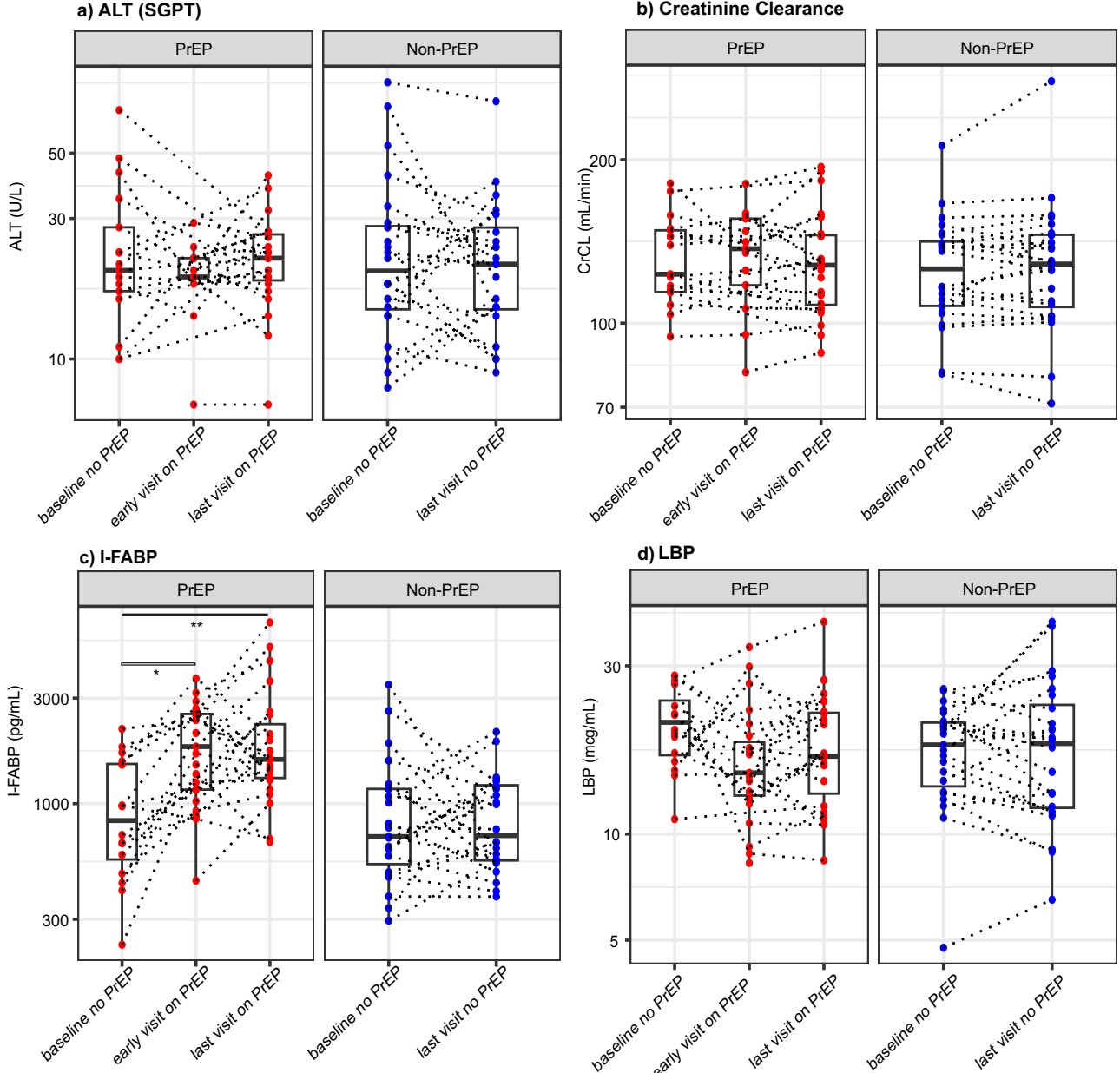

**Fig. 6 | Individual-level markers of PrEP side effects in liver, kidney, and intestine among PrEP and non-PrEP users. a** Hepatic marker ALT (U/L); **b** Renal filtration estimate creatinine clearance CrCl (min/mL); **c** Intestinal epithelial permeability marker I-FABP (pg/mL); and **d** Lipopolysaccharide Binding Protein LBP (mcg/mL). For ALT and CrCl, the sample sizes for each of the boxplots from left to right are $n = 17, 13, 23, 24$, and 24. For I-FABP and LBP, the sample sizes are $n = 16, 23, 23, 23$, and 24. The "early visit on PrEP" timepoint is the earliest study visit after the first evidence of PrEP use based on self-report and DBS. The "last visit on PrEP" visit is the last study visit with evidence of PrEP use, typically at the Week 72 visit for the 10th infusion. The mid-line of the box denotes the median and the ends of the box denote the 25th and 75th percentiles. The whiskers at the top and bottom of the box extend to the most extreme data points that are no more than 1.5 times the interquartile range (i.e., height of the box) or if no value meets this criterion, to the data extremes. Statistical comparisons were based on two-sided Wilcoxon signed rank tests for paired data. For ALT, the two-sided $p$-values are 1.0 (baseline vs. early visit on PrEP) and 0.85 (baseline vs. last visit on PrEP) among PrEP users, and 0.45 among non-PrEP users; for CrCl, the two-sided $p$-values are 0.69 (baseline vs. early visit on PrEP) and 0.93 (baseline vs. last visit on PrEP) among PrEP users, and 0.60 among non-PrEP users; for I-FABP, the two-sided $p$-values are 0.04 (baseline vs. early visit on PrEP) and 0.001 (baseline vs. last visit on PrEP) among PrEP users, and 0.80 among non-PrEP users; and for LBP, the two-sided $p$-values are 0.14 (baseline vs. early visit on PrEP) and 0.16 (baseline vs. last visit on PrEP) among PrEP users, and 0.82 among non-PrEP users. *denotes $0.01 \le p$-value < 0.05; **denotes $0.001 \le p$-value < 0.01; ***denotes $p$-value < 0.001.

impact on the pharmacokinetics of an HIV-1 monoclonal antibody and we propose a potential mechanism associated with increased intestinal epithelium permeability.

These findings have several important implications. First, given that oral PrEP has been licensed in several countries as an HIV-1 prevention tool, additional studies of mAbs for HIV prevention may benefit from including PrEP users in their PK measurements, to ensure optimal dosing in both PrEP and non-PrEP user populations. Given the long duration of I-FABP upregulation, it may also be important to assess if the increased clearance rate also affects other populations taking anti-retrovirals, such as people living with HIV, who participate in treatment interruption trials using monoclonal antibodies. Nevertheless, given the relatively low to moderate influence on VRC01 clearance (15%) and area under the curve (14%) between PrEP and non-

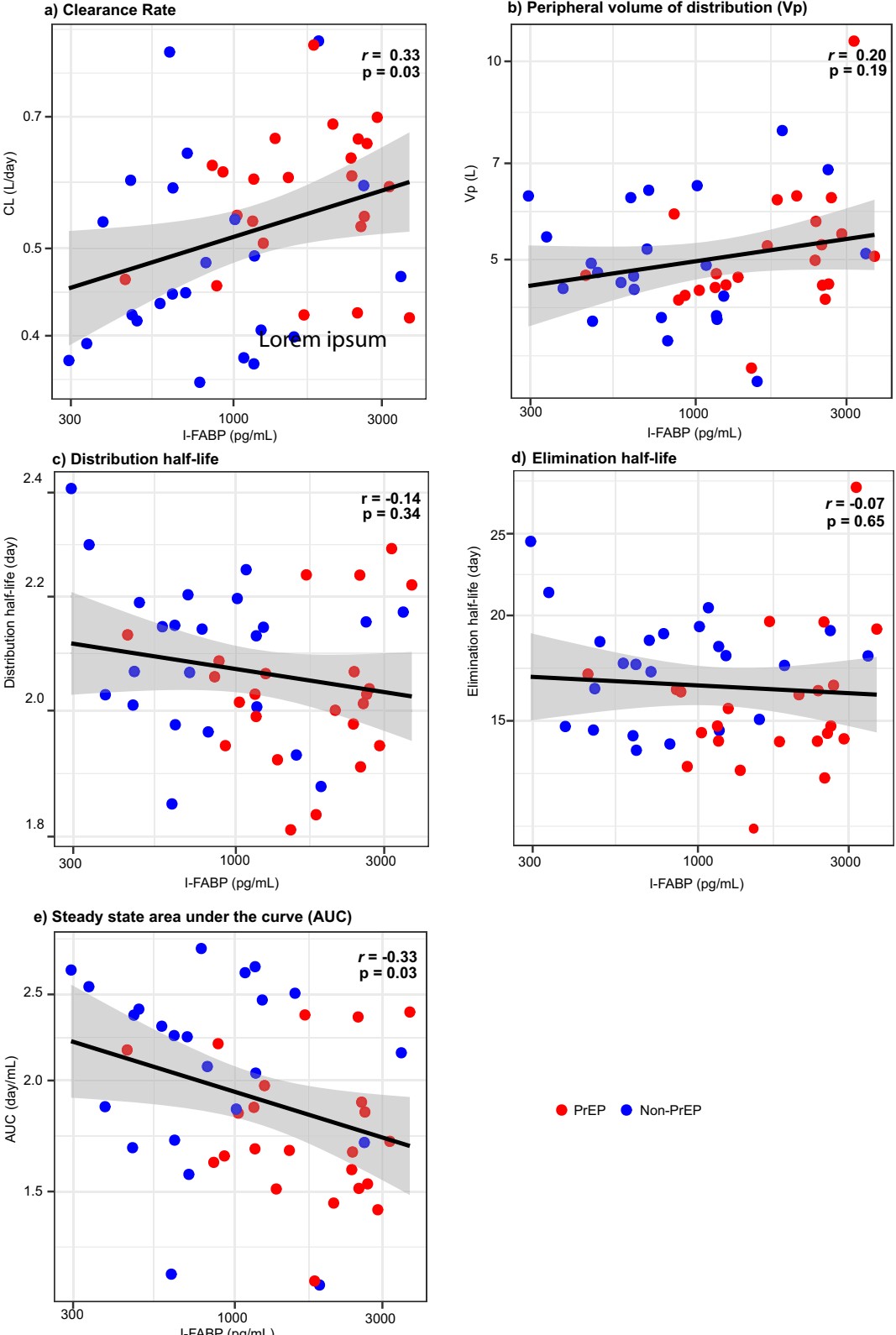

**Fig. 7 | Individual-level PK parameter estimates. a** clearance (CL) (L/day); **b** Vp (L); **c** Distribution half-life (day); **d** Elimination half-life (day); and **e** area under the curve (AUC) (day/mL) vs. median-scaled intestinal epithelial permeability marker I-FABP levels (pg/mL) based on the popPK model restricted to VRC01 serum concentration data collected after the "first visit on PrEP" timepoint. Data from 23 PrEP users and 23 non-PrEP users were included due to missing I-FABP level from one PrEP user and one non-PrEP user. Spearman correlation coefficients ($r$) and $p$-values for testing a non-zero correlation are included. A linear regression line with pointwise 95% confidence interval band (shaded area) is also included.

PrEP users, it is unlikely that dose adjustment would be needed for VRC01 had it been a licensed mAb for immuno-prophylaxis. However, this pattern will need to be verified for other HIV-1 mAbs, including LS mAbs designed for enhanced mucosal (re-)circulation, entering the prophylaxis pipeline. Similarly, because the PK of VRC01 resembles the PK of other IgG-based monoclonals, our investigation suggests that the PK of prophylactic or therapeutic antibodies for other diseases in PrEP users should be investigated to determine if optimal doses of immunotherapies are affected by PrEP.

## Methods

### Study overview
The data generation process and analysis steps of this study are summarized in Supplementary Fig. 11. Briefly, PrEP, and non-PrEP users were selected from AMP (MSM/TG) participants with and without evidence of PrEP use. Serum concentrations of VRC01 after each of the ten 8-weekly VRC01 infusions were measured by ELISA approximately every 4 weeks in these participants. Population pharmacokinetics (popPK) models were applied to analyze the concentration-over-time data and the Targeted Maximum Likelihood Estimation (TMLE) method was used to evaluate the impact of PrEP use on VRC01 PK, adjusted for potential confounding factors. Proinflammatory markers and markers associated with PrEP side effects among these participants were examined over time to aid interpretation of the study results.

### Clinical trial
Briefly, HVTN 704/HPTN 085 (ClinicalTrials.gov #NCT02716675) enrolled 2699 participants assigned male sex at birth or transgender individuals who have sex with men in Brazil, Peru and Switzerland and the US. Participants were randomized (1:1:1) to receive ten 8-weekly infusions of 10 mg/kg VRC01, 30 mg/kg VRC01, or placebo. All participants were offered HIV-1 oral PrEP (TDF-FTC) free of charge, with different timing of initiation and duration of adherence. During the study, participants donated blood collected in SST tubes for serum purification. For safety, creatinine and ALT levels in serum were monitored as safety laboratory measurements at every infusion timepoint. More information on the study can be found in Corey et al.[13]. The analyses in this manuscript are post-hoc.

### Study cohort
In this current study of PrEP effect on VRC01 pharmacokinetics, we randomly sampled a total of 234 participants from US-based HVTN 704/HPTN 085 sites who reached the week 88 study visit (16 weeks after the 10th and last VRC01 infusion) without HIV-1 and who did not permanently discontinue infusions during trial follow up. Participants were eligible for sampling irrespective of the number of VRC01 infusions received or the timing of infusions. Among these 234 participants, 77 did not self-report PrEP use or access the PrEP referral program, and 157 self-reported PrEP use during the AMP study. All available DBS samples collected at infusion visits of these self-reported PrEP users and self-reported non-PrEP users were measured and the results, one per visit, were included in the definition of PrEP users and non-PrEP users. A participant was defined as a PrEP user if they met all of the following criteria during the study: (1) accessed the PrEP referral program on at least one occasion per self-report, (2) self-reported intermittent or continuous PrEP use, and (3) confirmation of ≥3 positive TDF-FTC detection tests out of DBS samples collected at infusion visits. A participant was defined as a non-PrEP user if they did not self-report any PrEP use and had no DBS samples tested positive for TDF[14]. See more details on the DBS test below. Subsequently, a total of 24 PrEP users out of 31 eligible PrEP users and 24 non-PrEP users out of 32 eligible non-PrEP users were sampled, with an equal split in the low (10 mg/kg) and high (30 mg/kg) VRC01 dose groups.

### Dried blood spot (DBS) assay
The DBS assay used liquid chromatography and mass spectrometry to measure the levels of two pre-exposure prophylaxis (PrEP) drug anabolites, intraerythrocytic TFV-DP (tenofovir diphosphate) and FTC-TP (emtricitabine diphosphate) in DBS[37,83]. For drug level testing, 25 µL of blood from EDTA tubes was spotted five times onto 903 Protein Saver Cards (Whatman/GE Healthcare, Piscataway, NJ) (125 µL blood used in total). After spotting, the cards were dried for at least 2 h then stored at −80 °C prior to analysis and shipped on dry ice to the lab for assay. For analysis, a 3-mm diameter disk was punched from the blood spot on the card, using a micropuncher, followed by extraction with methanol:water and purification by solid phase extraction. Detectable concentrations used in the study were above the assay lower limit of detection of 31.25 fmol/punch and 0.125 pmol/punch for TFV-DP and FTC-TP, respectively. We used the concentration of TFV-DP only in subsequent analyses. More details are provided in Supplementary Methods.

### ELISA pharmacokinetics (PK) assay
VRC01 concentrations in serum samples collected from all available timepoints post VRC01 administration (including out of window visits, if any) through to the week 88 study visit were measured by an enzyme-linked immunosorbent assay (ELISA)[12,84]. Assays were blinded to PrEP status. Specifically, VRC01 concentrations in participant sera were quantified in 96-well plates on a Beckman Biomek-based automation platform according to the VRC/NVITAL standard operating procedure "5500-Automated ELISA on SCARA Core System." The monoclonal antibody 5C9 was coated onto Immulon-4HXB microtiter plates overnight at 4 °C at a concentration of 3.5 µg/mL. Plates were washed and blocked (10% FBS in PBS) for 2 h at room temperature. Duplicate serial 3-fold dilutions covering the range of 100–24,300 of the test sample were incubated 2 h at 37 °C followed by Horseradish Peroxidase - labeled goat anti-human antibody (1 h, 37 °C) and tetramethylbenzidine substrate (15 min, room temperature). Color development was stopped by addition of sulfuric acid and plates were read within 30 min at 450 nm via the Molecular Devices Paradigm plate reader. Final sample concentrations were based upon dilution corrected concentrations based upon linear regression of the standard curve covering the range of 5–125 ng/mL. Concentration values below the lower limit of quantification (=1.0 µg/mL) were replaced by 0.5 µg/mL in all calculations. If there were consecutive measurements below the limit, only the first one was included in the PK modeling. Additional experiments were also performed to verify lack of interference between TFV/FTC and VRC01 in the ELISA assay[85] (Supplementary Methods, Supplementary Table 8).

### Complete blood count (CBC), creatinine clearance (CrCl), and alkaline aminotransferase (ALT)
CBC with differential, creatinine, and ALT levels in serum as safety laboratory monitoring of VRC01 were collected at each VRC01 infusion visit and assayed at clinical diagnostic laboratories associated with each clinical site. Age, body weight and sex assigned at birth were used to calculate creatine clearance (CrCl) using the Cockcroft-Gault equation.

### Multiplex inflammation marker assay
Proinflammatory markers were measured in 1:2 diluted serum, in duplicate. The Meso Scale Discovery (MSD) V-Plex ProInflammatory Panel 1 kit (Catalog #: N05049A-1) was run according to manufacturer instructions. The assay monitors IFN-γ, IL-1β, IL-2, IL-4, IL-6, IL-8, IL-10, IL-12-p70, IL-13, TNF-α. Concentrations (pg/mL) of the different analytes were extrapolated from 4PL-fitted Calibrator Standard Curves (Catalog #: C0049-2) using the Mesoscale Discovery Workbench. Each plate run underwent quality control assessments to demonstrate less

than 30% CV among duplicates, and 70–130% recovery of calibrators and controls. Samples were rerun if the CV of the duplicates was higher than 30% CV. Most readouts for IL-1β, IL-2, IL-4, IL-12-p70, and IL-13 were below the limits of quantitation of the assays, and therefore excluded from the modeling. For analysis of IL-6, 6 measurements were below the plate-specific limits of quantitation and were replaced by half of the limits.

## Lipopolysaccharide (LPS) binding protein (LBP) ELISA

Serum samples were thawed on ice and centrifuged $10,000 \times g$ for 5 min before generating 1:800 dilutions to run in duplicates. The LBP ELISA kit was run according to manufacturer instructions (Antibodies-online Catalog number: ABIN5664982). Briefly, diluted samples, an LBP standard curve ranging from 50 to 1.5 ng/mL and a reference standard are plated on plates coated with an LBP antibody. After 1 h incubation in a shaker and washing the plate in the automated BioTek ELx405 plate washer, bound LBP was detected with a peroxidase-conjugated antibody specific for human LBP in a 1 h incubation in the shaker. Following the wash, bound conjugates were detected by the reaction of 3,3′,5,5′-Tetramethylbenzidine (TMB) for a 13 min incubation at room temperature protected from light, followed by $OD_{450}$ reading in the SpectraMax i3X ELISA plate reader. The standards were used to fit a 4PL curve from which all sample concentrations were extrapolated and adjusted for dilution. Duplicates with CVs higher than 30% were rerun to ensure accuracy.

## Intestinal fatty acid binding protein (I-FABP) ELISA

Serum samples were thawed on ice and centrifuged 10,000 g for 5 min before generating 1:5 dilutions to run in duplicates. The Human FABP2/I-FABP Immunoassay kit was run according to manufacturer instructions (R&D Systems, Inc., catalog number DFBP20). Briefly, diluted samples, an E. coli-expressed recombinant human I-FABP standard curve ranging from 1000 to 15.6 pg/mL and three reference standards are plated on plates coated with an I-FABP antibody. After 2 h incubation in a shaker and washing the plate in the automated BioTek ELx405 plate washer, bound I-FABP was detected with a peroxidase-conjugated antibody specific for human I-FABP in a 2 h incubation in the shaker. Following the wash, bound conjugates were detected by the reaction of 3,3′,5,5′-Tetra-methylbenzidine (TMB) for a 30 min incubation at room temperature protected from light, followed by $OD_{450}$ reading in the SpectraMax i3X ELISA plate reader. The standards were used to fit a 4PL curve from which all sample concentrations were extrapolated and adjusted for dilution. Duplicates with CVs higher than 30% were rerun to ensure accuracy.

## Cystatin C ELISA for GFR

Serum samples were thawed on ice and centrifuged 10,000 g for 5 min before generating 1:2000 dilutions to run in duplicates. The Human FABP2/I-FABP Immunoassay kit was run according to manufacturer instructions (Invitrogen, catalog # BMS2279). Briefly, diluted samples, a Cystatin C standard curve ranging from 3000 to 46.9 pg/mL and a control serum are plated on plates coated with a Cystatin C antibody. All the wells received a peroxidase-conjugated antibody specific for human Cystatin C and are incubated for 2 h in the shaker. Following the wash in the automated BioTek ELx405 plate washer, bound conjugates were detected by the reaction of 3,3′,5,5′-Tetramethylbenzidine (TMB) for a 20 min incubation at room temperature protected from light, followed by absorbance reading at 450 nm (for readout) and 630 nm (for reference) in the SpectraMax i3X ELISA plate reader. The standards were used to fit a 4PL curve from which all sample concentrations were extrapolated and adjusted for dilution. Duplicates with CVs higher than 20% were rerun to ensure accuracy[86]. Cystatin C concentrations were used to calculate GFR using the CDK-EPI Cystatin C equation.

## Anti-drug antibody (ADA) assay

ADAs were detected and characterized using a tiered testing strategy. In Tier I, a sensitive binding assay was used to determine if samples may have ADA present. In Tier II, the response was confirmed, typically by establishing the specificity of the response by competition with free drug. In Tier III, the response was characterized, typically with a neutralization reduction assay and/or a titering assay. For Tiers I and II as well as the titering assay, "bridging" assay formats are amongst the most common[87]. Specifically, a bridging assay to detect ADA against a biologic drug product began with covalently conjugating drug product with either biotin or the Sulfo-Tag label. Biotinylated and Sulfo-Tagged mAb were then combined and mixed with serum that may contain ADA. When an ADA response was present, complexes comprised of the ADA and one or both types of labeled mAb species could form. In some cases, because antibodies were multivalent, the ADA would act as a bridge between the biotinylated and Sulfo-Tagged drug in a ternary complex. Following an incubation to allow complex formation, this mixture was added to a proprietary streptavidin-functionalized plate, washed, and the presence of ADA responses, as indicated by the presence of ternary or "bridged" complexes, was detected. The Meso Scale Diagnostics (MSD™) instrument passed an electrical current through the plate, exciting any Sulfo-Tag within proximity of the plate surface, and resulting in electrochemiluminescence, which was expressed in relative light units (RLU). During assay qualification, a positivity cut point was established and utilized to determine whether a sample was ADA Screening Assay positive based on its RLU signal intensity. While only complexes containing both biotinylated and Sulfo-Tagged drug will result in signal, additional complexes are likely to form that are nonconducive to signal measurement (e.g., biotin:biotin or Sulfo-Tag:Sulfo-Tag bridged products). Formation of these nonproductive complexes has been taken into consideration during the development phase, and efforts have been made to maximize sensitivity by reducing the likelihood of formation of these nonproductive complexes. The criteria for establishing positivity cut points in the Tier I assay were designed to minimize the risk of false negatives. As such, a certain proportion of ADA negative subjects could be classified as positive in this screening assay but could later be investigated with Tier II confirmatory and, in some cases, Tier III characterization assays. The assays were all performed in 96-well plates for high throughput capacity. The screening assay has been qualified, and the specificity and titering assays have been deemed fit for purpose. Serum samples were assayed at a 1:12 dilution of serum for Tiers I and II. In Tier III, titers were defined as lowest fold dilution from a starting dilution of 1:12 at which the test sample remained above the assay cut point. When the average value of a test sample exceeded the assay cut point, that sample was classified as Tier I positive. Tier I positive samples were retested in the presence of free drug. If free drug reduced signal by more than the specificity threshold, these samples were considered Tier II confirmed. Test samples that were Tier II confirmed and had sufficient remaining volume were titered across a dilution series using the screening assay. The lowest fold dilution at which the test sample remained above the assay cut point was considered its titer. To ensure quality control, positive controls needed to exceed values established during qualification. If any control exceeded an average CV of 20%, an investigation of the run was performed. When any sample exceeded a CV of 20%, that sample was rerun.

## Population pharmacokinetics (PopPK) modeling

We analyzed individual concentration-time data using nonlinear mixed effects modeling with the Monolix software system (Version 2019R1) (https://lixoft.com/products/monolix/). The stochastic approximation of expectation-maximization (SAEM) method was applied to the modeling of the time-concentration data. VRC01 PK following IV administration was described by an open 2-compartment disposition model with first-order elimination from the central compartment. The

model was parameterized in terms of CL, Vc, Q, and Vp, denoting clearance from the central compartment (L/day), volume of the central compartment (L), inter-compartmental distribution clearance (L/day) and volume of the peripheral compartment (L). An exponential between-individual random effect was considered for CL and Vp based on patterns observed in the data.

Identification of baseline covariates predictive of PK variability was performed to better understand the sources of observed inter-individual variability in CL and Vp. The baseline covariates that were screened for this analysis were pre-defined, including dose group (10 mg/kg or 30 mg/kg), demographic variables: age (years), body weight (kg), race (to ensure body mass index (kg/m$^2$)), behavioral risk score;[16] clinical variables: pulse rate (beats/min), respiratory rate (breaths/min), diastolic blood pressure (mmHg), systolic blood pressure (mmHg), temperature (°C); safety lab variables: CrCl (mL/min), erythrocyte mean corpuscular volume (fL), ALT (U/L), hematocrit (%), hemoglobin (g/dL), platelets (10$^3$/mm$^3$), leukocyte count (10$^3$/mm$^3$), lymphocyte count (cells/mm$^3$), monocyte count (cells/mm$^3$), neutrophil count (cells/mm$^3$), basophil count (cells/mm$^3$), and eosinophil count (cells/mm$^3$); inflammatory marker values including IFN-γ, TNF-α, IL-6, IL-8, IL-10, as well as the time-varying PrEP use indicator (yes or no). The time-varying PrEP use indicator is defined at each infusion and mid-infusion visits as "yes" if both the PrEP drug concentration measured using DBS was detectable and the participant self-report PrEP use, and as "no" otherwise. Since PrEP concentration were only measured at infusion visits, not mid-infusion visits, PrEP concentrations at mid-infusion visits with self-reported PrEP use were imputed to be equal to the last PrEP concentration when participant self-reported being on PrEP. PrEP concentrations at mid-infusion visits with no self-reported PrEP use were imputed to be equal to the last PrEP concentration when participant self-reported being not on PrEP.

The above covariates were considered in subsequent analyses if they were correlated with either the individual-level CL or Vp estimates with the Spearman correlation coefficient ≥ 0.3 and $p$-value < 0.05 for testing a non-zero correlation.

**Comparison of PK features between PrEP and non-PrEP users**
Five individual-level PK features – CL, Vp, steady-state dose-normalized area under the time-concentration curve, distribution half-life, and elimination half-life were derived from the base popPK model without covariate adjustment. Given VRC01 concentrations were measured in serum samples collected at study visits subject to relatively large visit windows (−7 to 49 days for infusion visits and ±7 days for non-infusion visits other than Day 61), comparisons of PK profiles between PrEP and non-PrEP users were not performed on the observed visit-specific serum concentrations directly. Rather, such comparisons were based on participant-level PK parameters that capture key features of participant-specific PK profiles. These PK parameters were estimated from the base popPK model that characterizes participant-specific concentrations observed over time, without adjusting for any of the covariates that may predict PK among the 48 PrEP and non-PrEP users. For comparing non-randomized groups of interest, such as the PrEP vs. non-PrEP user groups, to reduce confounding bias the targeted minimum loss-based estimation (TMLE) method[28] was used to estimate the mean of each feature for each group, adjusted for potential predictors of PK variability: age, body weight, race (Black/African American vs. other racial identities), CrCl, behavioral risk score, IFN-γ and IL-10 (implemented in the *tmle* R package)[27]. TMLE is an alternative to standard linear or nonlinear regression that can improve robustness and efficiency. All TMLE estimation results of means were averaged over 20 runs with a fixed random seed on top of the 10-fold cross-validation estimation procedure to ensure stability of the estimates. The set of learning algorithms used by TMLE for estimating the mean outcome conditional on baseline covariates are SL.glm, SL.step, SL.ranger, SL.earth, SL.glmnet, and SL.mean[14]. In addition, to account

for variability and co-variability of the individual-level estimates for each PK feature due to the fact that they were derived from a common popPK model, a bootstrap procedure based on 500 datasets was used to calculate the empirical variances of the estimates for each group (after outliers were removed) and to derive the 95% confidence interval, as well as to test for a non-zero mean difference between the two groups. The Holm method[88] was used to adjust for multiple comparisons of the five PK features. A TMLE sensitivity analysis was also performed that excluded data from the single PrEP user (ID = 21) whose PK parameter estimates were unstable.

### Reporting summary
Further information on research design is available in the Nature Portfolio Reporting Summary linked to this article.

## Data availability
The source data supporting the findings of this study have been deposited in the figshare database under accession code: https://doi.org/10.6084/m9.figshare.23800698.

## Code availability
The code for the base population pharmacokinetics (popPK) model, the popPK Model via the Monolix software, and the R code that implemented the TMLE test supporting the analyses of this study are available in the figshare databased under accession code: https://doi.org/10.6084/m9.figshare.23800698.

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

## Acknowledgements

This work was supported by the National Institute of Allergy and Infectious Diseases (NIAID) U.S. Public Health Service Grants UM1 AI068614 [LOC: HIV Vaccine Trials Network] to L.C., UM1 AI068619 [LOC: HIV Prevention Trials Network] to M.S.C, UM1 AI068635 [HVTN SDMC] to P.B.G and Y.H., UM1 AI068617 [HPTN SDMC] to D.D., and UM1 AI068618 [HVTN Laboratory Center] to M.J.M. We thank Gilead for donating TDF-FTC (Truvada®) freely to participants in the AMP trials. We thank Erika Rudnicki, Nidhi Kochar, and Karan Shah for their data management and data

analysis support. We thank Greg Mize for conducting quality control of the ELISA runs. We thank Margarita M. Gomez Lorenzo and David Burns for their clinical monitoring expertise during the conduct of HVTN 704/ HPTN 085. We thank Julia Hutter, Will Hahn, Gail Broder for their feedback on early drafts. We thank Kwang Low and Leonid Serebryannyy for performing the experiments related to lack of interference between VRC01 concentrations and the presence of oral PrEP in the ELISA assay. We thank the HVTN 704/HPTN 085 study participants and the effort of the following 16 clinical sites (Principal Investigator), in alphabetic order, for recruiting participants included in this analysis: Atlanta - Emory University (Sri Edupuganti), Atlanta - Ponce de Leon (Carlos del Rio), Birmingham - Univ of Alabama (Paul Goepfert), Boston - Brigham and Women's (Lindsey Baden), Boston - Fenway Health (Ken Mayer), Chapel Hill (Cyndy Gay), Cleveland (Jeffrey Jacobson), Columbia - Bronx Prevention (Jessica Justman), Nashville (Spyros Kalams), New Jersey Medical School (Shobha Swaminathan), New York - Columbia University (Magdalena Sobieszczyk), NYBC Union Square (Hong Van Tieu), Philadelphia/U of Pennsylvania (Ian Frank), San Francisco General Hospital (Susan Buchbinder), Rochester (Michael Keefer), UW: FHCRC (Juliana McElrath). The content of this manuscript is solely the responsibility of the authors and does not necessarily represent the official views of the National Institutes of Health. The funders had no role in study design, data collection and analysis, decision to publish, or preparation of the manuscript.

## Author contributions

Conceptualization: Y.H., M.J., M.S.C., L.C, P.B.G., and M.P.L. Clinical supervision: S.K., P.A., E.S., S.E., and N.M. Laboratory supervision: M.P.L., R.K., M.E.A., P.L.A., J.H., and M.J.M. Laboratory data generation: E.M., Y.A., and J.W. Statistical analysis: Y.H., L.Z., and M.P.L. Data Interpretation: Y.H., L.G., and M.P.L. Funding acquisition: Y.H., D.D., R.K., M.J.M., M.S.C, L.C., and P.B.G. Writing—original draft: Y.H., M.P.L., and H.A. Writing—review and editing: All coauthors.

## Competing interests

The authors declare no competing interests.

## Ethical approval

We complied with all relevant ethical regulations in analyzing these data. For the AMP trial, central and site-specific institutional review boards and ethics committees reviewed and approved the initial protocol and each subsequent version. All participants provided written informed consent, and new consent was obtained for each version of the protocol. The ClinicalTrials.gov number is NCT02716675.

## Additional information

[1]Vaccine and Infectious Disease Division, Fred Hutchinson Cancer Center, Seattle, WA 98109, USA. [2]Department of Global Health, University of Washington, Seattle, WA 98196, USA. [3]Family Health International, Durham, NC 27710, USA. [4]Thayer School of Engineering, Dartmouth College, Hanover, NH 03755, USA. [5]Vaccine Research Program, Division of AIDS, National Institute of Allergy and Infectious Diseases, National Institutes of Health (NIH), Rockville, MD 46340, USA. [6]Department of Medicine, Division of Infectious Diseases, Emory University School of Medicine, Atlanta, GA 30322, USA. [7]University of Zimbabwe Clinical Trials Research Centre, Harare, Zimbabwe. [8]Vaccine Research Center (VRC), National Institute of Allergy and Infectious Diseases, National Institutes of Health, Bethesda, MD, USA. [9]Colorado Antiviral Pharmacology Laboratory and Department of Pharmaceutical Sciences, Skaggs School of Pharmacy and Pharmaceutical Sciences, University of Colorado-AMC, Aurora, CO 80045, USA. [10]Institute for Global Health and Infectious Diseases, University of North Carolina at Chapel Hill, Chapel Hill, NC, USA. [11]Departments of Medicine and Laboratory Medicine, University of Washington, Seattle, WA 98195, USA. [12]Department of Biostatistics, University of Washington, Seattle, WA 98195, USA. [13]These authors contributed equally: Yunda Huang, Maria P. Lemos. ✉e-mail: yunda@fredhutch.org

