## [Peer Review File · Nature Communications]

Adults on Pre-Exposure Prophylaxis (tenofovir-emtricitabine)
Have Faster Clearance of Anti-HIV Monoclonal Antibody
VRC01Reviewers' Comments:

Reviewer #1:

Remarks to the Author:

The manuscript by Huang Y et al, addresses an important question in the field of HIV prophylaxis on the potential interaction of newly developed monoclonal antibodies (i.e., VRC01) when used concomitantly with oral PrEP (tenofovir-emtricitabine), the standard and widely utilized HIV prophylactic approach. Despite the small n value, this study has identified an interesting finding i.e., an alteration of the pharmacokinetic (PK) properties of VRC01 when administered with PrEP, and further identified a PrEP induced increased intestinal permeability as a potential underlying mechanism explaining the changes in VRC01 PK. Extensive work and careful analysis of the data have been applied. Comments below are provided for manuscript improvement.

Major Comments:

- 1) The authors should clearly address the rationale of combining these newly developed monoclonal antibodies with PrEP since oral prophylactic therapy with tenofovir and emtricitabine has been highly effective globally. What are the clinical advantages of this combination? Could the monoclonal antibodies be used as a single prophylactic measure? This will have the clinical advantage of avoiding the need for oral PrEP and associated adherence issues.
- 2) The study identified an increased in intestinal epithelial permeability, potentially induced by PrEP administration. Although several criteria have been selected for the subjects' enrollment, the variability of PrEP use /intake is of some concern in this study. For example, not all the subjects used PrEP continuously, some subjects were already on PrEP at the time of the study initiation while others started prophylactic treatment later on, serum concentrations of PrEP (TDF/FTC) during the study period were not measured.
- 3) This study didn't report any serum concentrations of TDF/FTC, yet has documented an increased intestinal permeability due PrEP. As PrEP drugs are primarily absorbed through the small intestine, do we anticipate that the induced increased permeability could alter the absorption of PrEP? Or the PK of PrEP drugs? This could potentially result in altered efficacy however it has not been clearly identified in the clinic.
- 4) The authors should further elaborate on the pre-clinical and clinical evidence that PrEP can alter intestinal permeability. Is it known if tight junction proteins of the intestinal epithelial cells are dysregulated by PrEP? These could be additional markers that could be examined.

Minor Comments:

- 1) When addressing processes such as the ones mediated by the cytochrome CYP450 (lines 77, lines 327-328), the authors should review their statement indicating that this is "drug transporter pathway", rather these processes are metabolizing pathways or processes that result in drug or mAbs metabolism or degradation. Drug transport is the process which mediates permeability of drugs across biological membranes either by simple diffusion (membrane pore) and /or by a specific process involving a membrane-associated transporter.
- 2) Line 76 "ADA" should be spelled out. This acronym is spelled later on, line 211

Reviewer #2:

Remarks to the Author:

This is a well-written, important manuscript evaluating the effect of oral TDF/FTC PrEP on the PK of

VRC01, an HIV-1 broadly neutralizing antibody that has been evaluated for preventive efficacy in the AMP study. Noteworthy results include the finding that VRC01 recipients who took PrEP had a significantly faster VRC01 clearance rate and lower VRC01 AUC compared to non-PrEP users – these effects are predicted to result in a 14% lower VRC01-mediated preventive efficacy among PrEP users. Additionally, the research team have done a thorough evaluation of potential mechanisms of these effects, and found that levels of I-FABP were elevated among PrEP users, suggesting increased epithelial intestinal permeability, and these elevated levels were positively associated with higher VRC01 clearance. These findings are highly significant, as this is the first study demonstrating an effect of oral TDF/FTC PrEP on the PK of a monoclonal antibody, and has important implications which are outlined in the discussion, including the need to assess PK of additional mAbs being tested for HIV prevention among PrEP users vs. non-users; the importance of assessing whether similar findings are seen in people living with HIV who are on antiretrovirals and participate in a treatment interruption trial using mAbs, and finally evaluating whether the PK of other prophylactic or therapeutic antibodies for other diseases in PrEP users is impacted.

The conclusions in this paper are well supported by the detailed methods and analyses presented in the manuscript. The authors point out that urine samples were not collected as part of AMP, and therefore they were not able to explore directly any tubular secretion or reabsorption detects of the mAb. Additionally, due to the large size the AMP study, they did not collect any mucosal samples to allow exploration of VRC01 clearance in intestinal secretions or mechanisms of intestinal pathology. In the discussion, the authors point to 2 possible mechanisms for how PrEP could affect VRC01 transport in the intestinal epithelium – affecting turnover of intestinal epithelial cells and/or impacting the endosomal transport and recycling of antibodies. It would strengthen the discussion if the authors could briefly state what additional studies are recommended to differentiate these two mechanisms in future studies.

The methodology used in this study are well-described and in general are sufficient to be reproduced. Under the dried blood spot assay description, it would be helpful to note whether samples were frozen at sites and within what time window after sample collection. Also, it would be helpful to specify the lower limit of detection for TFV-DP in DBS here in the text.

Here are some additional minor recommendations / comments for clarity:

Line 46 – AWARE trial showed that risk reduction counseling was not effective in reducing STIs - consider removing risk reduction counseling from the text

Line 76- define ADA

Line 94 – would be helpful to specify the level of PrEP uptake and adherence in the AMP study here in the text

Line 154 – change to creatinine clearance (not creatine clearance)

Line 226-227 The sentence “since creatinine clearance...” seems incomplete

Line 480 – Should “Samples were rerun in the CV of the duplicates was higher than 30% CV” be changed to “Samples were rerun if the CV of the duplicates was higher than 30% CV”?

Reviewer #3:

Remarks to the Author:

This study sought to understand why the clearance of the mAb VRC01 was faster in subjects on oral pre-exposure prophylaxis (PrEP) consisting of tenofovir and emtricitabine compared to subjects not on PrEP. There is no clear or obvious pharmacokinetic interaction, hence the novel science being conducted here is worthy of reporting. However, there are several items that need clarification or revision.

Specific Comments

Abstract

- third sentence, check units on "dose-normalized VRC01 serum concentrations". 0.29 day/mL seems wrong. It should be something like 0.29 ug/mL/day, e.g.
- Just because the clearance was 14% faster and serum concentrations 14% lower doesn't always mean that efficacy will be 14% lower too. It depends on the receptor occupancy of VRC01 with its target receptor, the expression levels of that target receptor in patients with HIV-1, the disease burden of each individual, etc. Given that these are healthy subjects without HIV, could there be another physiological reason for inter-individual variability in the target receptor (CD4 binding site)? I understand the PT80 values were calculated to assess efficacy, but again, that metric incorporates the serum concentration.

Introduction

- More background information is needed about VRC01 itself and the mechanism by which it is believed to prevent HIV infection upon exposure. Simply stating that it is a CD4-binding site targeting mAb for prophylactic HIV prevention doesn't mean much to the lay reader. A brief 2-3 sentence background on why inhibiting (I'm assuming VRC01 binds CD4 and inhibits this site from interacting with some other ligand) is believed (with citations) to prevent HIV infection and/or transmission.
- Second paragraph...the dosing schedule of every 8 weeks, how was that optimized, but the dose amount (10 mg/kg or 30 mg/kg) not? Please cite the study(s) that selected 8 weeks as the optimal dosing interval. This prolonged schedule is atypical of most therapeutic mAbs and in particular with VRC01 having a known half-life of 15 days.
- Second paragraph, you state "...viruses that were sensitive to VRC01." Is it known if certain strains of HIV are more or less sensitive to VRC01, presumably based on a yet-to-be-stated mechanism of action?

Results

- "Markers of Hepatic and Renal..."; third paragraph, you state faster clearance of VRC01 were not "temporarily" linked. Do you mean "temporally"?
- "Impact of intestinal permeability on VRC01 PK", second sentence: "...at this time-point, only among PrEP users (not non-PrEP users)". did you mean to say "...(not non-PrEP users)"?
- Was the impact of serum albumin concentrations assessed for correlation with VRC01 clearance? There is plenty of literature on the correlation of albumin and mAb clearance due to increased general protein turnover/catabolism.

Methods

- More information is needed about the LC-MS/MS assays used to measure TDF, TFV and emtricitabine (FTC), and the metabolites TFV-dp and FTC-TP. Was a single LC-MS/MS assay used to measure all analytes, or because two DBS punches were used, were there two separate assays? More information is needed about this...

Table 2: Why does AUC have units of "Days/mL"?

Table 2/Figure 3:Vp in Table 2, using TMLE-predicted, covariate-adjusted mean values, had a nonsignificant difference between PrEP vs non-PrEP users (5.24 L vs 4.88 L). However, in Figure 3b, the visualization of this is drastic and presents as a clearly distinct profile.

Point-by-Point Response to Reviewers' Comments

Reviewer #1 (Remarks to the Author):

The manuscript by Huang Y et al, addresses an important question in the field of HIV prophylaxis on the potential interaction of newly developed monoclonal antibodies (i.e., VRC01) when used concomitantly with oral PrEP (tenofovir-emtricitabine), the standard and widely utilized HIV prophylactic approach. Despite the small n value, this study has identified an interesting finding i.e., an alteration of the pharmacokinetic (PK) properties of VRC01 when administered with PrEP, and further identified a PrEP induced increased intestinal permeability as a potential underlying mechanism explaining the changes in VRC01 PK. Extensive work and careful analysis of the data have been applied. Comments below are provided for manuscript improvement.

Major Comments:

1) The authors should clearly address the rationale of combining these newly developed monoclonal antibodies with PrEP since oral prophylactic therapy with tenofovir and emtricitabine has been highly effective globally. What are the clinical advantages of this combination? Could the monoclonal antibodies be used as a single prophylactic measure? This will have the clinical advantage of avoiding the need for oral PrEP and associated adherence issues.

Response: Thank you for bringing up this important question. We agree with the reviewer that there are potential clinical advantages of concomitant use of oral PrEP and mAb such as leveraging the longer half-life of mAbs to provide protection especially during missed daily oral doses of tenofovir-emtricitabine. However, at this point, monoclonal antibodies for HIV immunoprophylaxis have demonstrated only partial efficacy when used alone; they do not yet accomplish sufficient protection to merit direct comparisons with PrEP usage. They are not yet licensed for use alone or as a backup in case of PrEP adherence issues.

The context of the research described in this paper was not to evaluate the clinical benefit of concomitant use of PrEP and mAb. Rather, as our reviewer pointed out, the primary objective of the AMP trials was to evaluate the safety and efficacy of the monoclonal antibody VRC01 in the background of the standard use of other prevention modalities, including condoms and PrEP etc.. The reason why there were some AMP trial participants who had concomitant use of VRC01 and oral PrEP in the AMP study was because oral PrEP was offered as part of the risk reduction package to all AMP participants free of charge, after they were randomly assigned to receiving VRC01 or placebo. The uptake of PrEP was voluntary in these AMP trial participants. However, data collected from these AMP trial participants who had concomitant use of PrEP and VRC01, by choice not by design, were scientifically unique and provided the basis for the investigation of potential interactions between PrEP and VRC01, as described in our paper. In this paper, based on data from a subset of AMP trial participants who were on concomitant use of oral PrEP and mAb, we investigated the effect of PrEP on the pharmacokinetics profile of VRC01.

To address this important comment, we clarified in the following sentence in the Introduction that passive immunization of mAbs provides a new potential prevention modality, either alone or together with other existing modalities:

“Passive immunization with broadly neutralizing monoclonal antibodies (mAbs) provides a novel approach as an additional HIV-1 prevention modality, alone or in combination with other existing prevention modalities.”

And in the second paragraph of the Introduction, we added:

“Therefore, although concomitant use of PrEP and VRC01 was only observed in a subset of AMP participants given the voluntary uptake of PrEP.”

2) The study identified an increased in intestinal epithelial permeability, potentially induced by PrEP administration. Although several criteria have been selected for the subjects' enrollment, the variability of PrEP use /intake is of some concern in this study. For example, not all the subjects used PrEP continuously, some subjects were already on PrEP at the time of the study initiation while others started prophylactic treatment later on, serum concentrations of PrEP (TDF/FTC) during the study period were not measured.

Response: Our reviewer is correct that there is variability of PrEP use/uptake in the study cohort. This is expected in studies like this because the start and stop of PrEP use was voluntary in the AMP study, offered as part of the HIV prevention toolkit. In addition, requiring consistent compliant to a daily use of PrEP for two years would be challenging for prevention purposes; compliant was variable even in the two clinical trials of PrEP that provided efficacy data to support its licensure (Grant et al., 2010; Baeten et al., 2012).

However, this is not to suggest that we could ignore the variability of PrEP use/uptake in our study. To address the concern of different start/stop/resume time of PrEP use across individuals, we specifically included a supportive analysis that accounted for each individual's PrEP on/off status over each time-point. The results were described in the effect of PrEP use on VRC01 PK in the original submission. The relevant text is pasted below for convenience.

“Lastly, since PrEP use could be intermittent for some of the PrEP users as shown in Figure 1, additional analyses were performed to estimate the effect of current PrEP use on VRC01 PK that accounted for the time-varying status of PrEP use indicated by TDF-detectable DBS testing results (Supplementary Tables 6 & 7, Supplementary Figs. 7 & 8). The same trend was observed, showing a clearance rate that is 1.02 (95% CI: 1.02, 1.03) fold higher when PrEP was detected (vs. not detected).”

To address the concern that some individuals were already on PrEP at the time of AMP study enrollment, we included the analysis of using a “new baseline” where 7 PrEP users who started PrEP prior to study enrollment were not considered in this new baseline but at the later post-baseline time-point. The results are described in the paper and in Figures 5 & 6, as well as Supplementary Figure 9 and Supplementary Tables 1 & 5 in the original submission. The text referring to this is partially pasted below for convenience.

“The “early visit on PrEP” time-point was defined as the earliest study visit after the first evidence of PrEP use based on self-report and DBS. For the 17 PrEP users who had no evidence of PrEP use prior to enrollment, this “early visit on PrEP” is ~ 4 weeks after the first evidence of PrEP use during the AMP study; for the 7 PrEP users who had evidence of PrEP use prior to enrollment, this “early visit on PrEP” is the enrollment visit. The “last visit on PrEP” visit is the last study visit with evidence of PrEP use, typically at the Week 72 visit for the 10th infusion, and a median of 439 days (IQR 233-546 days) since the self-reported date of PrEP uptake.”

In addition, to address the concern raised by our reviewer that “serum concentrations of PrEP (TDF/FTC) during the study period were not measured”, we clarified in the description of the DBS assay that we used the concentration of TFV-DP in DBS to define PrEP use status. This is because

TFV-DP accumulates appreciably in red blood cells with repeated daily doses and has a half-life of 17 days in this cellular compartment (Castillo-Mancilla, JR 2013). Therefore, measurement of TFV-DP in DBS is useful to assess cumulative patterns of adherence over 6-8 weeks. Given that the half-life of VRC01 in serum is about 15 days, we believe it is relevant to monitor the use of PrEP using DBS and assess its effect on the PK of VRC01. In the Results under ‘study population and baseline characteristics’, we also added:

“Particularly, DBS samples were used to measure red blood cell concentrations of the PrEP metabolites emtricitabine-triphosphate (FTC-TP) and tenofovir diphosphate (TFV-DP) concentrations representing a combination of recent and cumulative dosing of PrEP, respectively.”²⁶

We clarified the DBS Assay section of our methods to read:

“As described previously,^{37,83} the DBS assay used liquid chromatography and mass spectrometry to measure the levels of two pre-exposure prophylaxis (PrEP) drug anabolites, intraerythrocytic TFV-DP (tenofovir diphosphate) and FTC-TP (emtricitabine diphosphate) in DBS. For drug level testing, 25 ul of blood from EDTA tubes was spotted five times onto 903 Protein Saver Cards (Whatman/GE Healthcare, Piscataway, NJ) (125 ul blood used in total). After spotting, the cards were dried for at least 2 hours then stored at -80C prior to analysis and shipped on dry ice to the lab for assay. For analysis, a 3-mm diameter disk was punched from the blood spot on the card, using a micropuncher, followed by extraction with methanol:water and purification by solid phase extraction. Detectable concentrations used in the study were above the assay lower limit of detection of 31.25 fmol/punch and 0.125 pmol/punch for TFV-DP and FTC-TP, respectively. We used the concentration of TFV-DP only in subsequent analyses.”

Also, we added the following text to “DBS Assay” in the Supplementary Materials:

“The methods described here are specific to testing for two PrEP drug anabolites, intraerythrocytic TFV-DP (tenofovir diphosphate) and FTC-TP (emtricitabine diphosphate).^{1,2} TFV-DP is the active metabolite exists within cells after TDF is metabolized, and is responsible for inhibiting viral replication within infected cells; FTC-TP is the active intracellular form generated after FTC is metabolized within cells, and contributes to the overall inhibition of HIV replication via incorporation into viral DNA.

For drug concentration testing, 25 ul of blood from EDTA tubes is spotted five times onto 903 Protein Saver Cards (Whatman/GE Healthcare, Piscataway, NJ) (125 ul blood used in total). After spotting, the cards are dried for at least 2 hours and then placed in plastic bags and stored in a sample box with desiccant and humidity indicators. For short term storage, room temperature (<5 days) or 4C may be used; for longer term storage, -20C and -80C have been shown to have acceptable stability up to 5 years (reference 6.2, validation report).

For the extraction of analytes from DBS, a 3-mm diameter disk is punched from the blood spot on the card, using a micropuncher. A punch from a clean Protein Saver Card is performed in between each DBS sample in order to avoid analyte contamination from the previous DBS punch. The disk is placed in a microcentrifuge tube with 500 ul of 70:30 methanol-water solution. Following extraction, the lysed cellular matrix is subjected to solid phase extraction procedures to prepare the sample for liquid-chromatography tandem mass spectrometry. The validated quantifiable linear range for TFV-DP is 25–2000 fmol/sample and that for FTC-TP is 0.1–200 pmol/sample for a 3-mm punch. Because 400ul of the 500ul are assayed

(comprising the “sample”), the lower limit of quantification of TFV-DP for a 3mm punch is 31.25 fmol/punch and that for FTC-TP is 0.125 pmol/punch. Stable labeled isotopic internal standards are used to ensure accuracy and precision in various cell matrices.

As previously reported, TFV-DP accumulates appreciably in red blood cells with repeated daily doses and has a long half-life of 17 days in this cellular compartment. Therefore, measurement of TFV-DP in DBS is useful to assess cumulative patterns of adherence over longer periods, and provides valuable insight into both dosing recency and cumulative doses from variable adherence patterns.^{3,4} In all subsequent analyses, we defined PrEP use status based on TFV-DP concentration in DBS because the decay kinetics and levels of effective use have been reasonably well established for TFV-DP.

Lastly, the variability of PrEP adherence in the study participants was in fact helpful to examine the association between I-FABP levels in serum at the “last confirmed PrEP use” timepoint and the cumulative TFV-DP accumulation in the red blood cells at that timepoint. Please see our response to the next question.

3) This study didn't report any serum concentrations of TDF/FTC, yet has documented an increased intestinal permeability due PrEP. As PrEP drugs are primarily absorbed through the small intestine, do we anticipate that the induced increased permeability could alter the absorption of PrEP? Or the PK of PrEP drugs? This could potentially result in altered efficacy however it has not been clearly identified in the clinic.

Response: Thanks for raising this interesting question. We quantitated PrEP usage based on the dry blood spots, because as now clarified in the results:

“Particularly, DBS samples were used to measure red blood cell concentrations of the PrEP metabolites emtricitabine-triphosphate (FTC-TP) and tenofovir diphosphate (TFV-DP) concentrations representing a combination of recent and cumulative dosing of PrEP, respectively²⁶

Castillo-Mancilla et al 2013 showed an excellent correlation between FTC in plasma and FTC in DBS. Therefore, we believe defining PrEP use status based on DBS (monitoring months of usage) as opposed to serum/plasma concentrations of TFV-DP (monitoring daily uptake) is operationally advantageous in our setting, that followed participants for up to 2 years.

The IPREX study demonstrated that specific concentrations per punch in DBS correspond to a certain number of PrEP doses per week and established an effective-use threshold of 700 fmoles/punch of TFV-DP. We have now reported such data in Results:

“Among these PrEP users, at the last visit with confirmed PrEP usage, their DBS samples contained a median TFV-DP level of 1145 fmol/punch (range 540-2437), suggesting an average of at least 4 doses/week in the past 6-8 weeks.²⁶”

To address the reviewer's question regarding PrEP absorption, we assessed the correlation between I-FABP levels and TFV-DP concentrations in DBS samples at the last visit on PrEP among PrEP users, and have now added the corresponding figure as Supplementary Figure 10 (upper panel, Figure below). We found that they are positively correlated with a Spearman correlation coefficient of 0.61. This result suggests that PrEP can achieve effective concentrations (TFV-DP > 700 fmol/punch) in blood at timepoints when individuals have high I-FABP. Similar correlation was also seen between FTC-TP concentration and I-FABP level at the same timepoint (bottom panel, Figure below). Thus, although further research is needed to define accurately the impact of

intestinal permeability on the absorption of PrEP, ideally with sampling in serum within hours of administration, we found no evidence of reduced absorption of PrEP in individuals with evidence of high intestinal permeability based on our current data.

Lastly, the results of LBP also suggest that the enhanced clearance of VRC01 is selective and does not completely compromise intestinal transport mechanisms. This was addressed in the discussion as follows:

“Consistent with the lack of other pro-inflammatory markers, LBP was not up regulated after either ~1 month or ~14 months of reported PrEP usage, suggesting that pathways mediating intestinal bacterial LPS permeability, were not affected by PrEP. This result highlights that the intestinal permeability changes observed among PrEP users may be selective – TDF-FTC may alter specific transport or catabolic functions of intestinal epithelial cells but may preserve intact paracellular barriers⁶³ and endocytosis of lipid rafts^{68,69} that have been shown to regulate intestinal LPS entry.”

4) The authors should further elaborate on the pre-clinical and clinical evidence that PrEP can alter

intestinal permeability. Is it known if tight junction proteins of the intestinal epithelial cells are dysregulated by PrEP? These could be additional markers that could be examined.

Response: At the request of the reviewer, we have added the following paragraph to the subsection on intestinal permeability over time between PrEP and non-PrEP users, to discuss the known literature regarding intestinal side effects of PrEP:

“GI side effects among oral PrEP users⁵³⁻⁵⁵ include nausea, vomiting, diarrhea, stomach pain, and unintended weight loss. These are commonly referred to as “PrEP Startup Syndrome,” which occur more frequently within the first month after PrEP start, and tend to resolve on their own.⁴⁶ The exact mechanism causing these is unknown, but PK studies in animal and in vitro models have demonstrated that TDF-FTC accumulates in the intestine at higher concentrations than blood.⁵⁶”

In the discussion of the original submission, we also addressed potential mechanisms for PrEP’s selective effects on intestinal permeability such as the shortening of the intestinal surface (which has been shown for protease inhibitors), or alterations to FcRn recycling:

“On one hand, PrEP could be affecting the dynamic turnover of intestinal epithelial cells,^{74,75} shortening the villi, and reducing the absorptive surface area, where antibody is recycled from the lumen. This kind of absorptive effect has been demonstrated in mouse models for nelfinavir (NFV), indinavir (IDV), didanosine (DDI) and zidovudine (AZT),⁷⁶ and could have implications for the absorption of nutritional contents and other medications taken orally. On the other hand, PrEP could be modifying the endosomal transport and recycling of antibodies, primarily mediated by the neonatal Fc receptor (FcRn), which rescues up to 2/3 of the endogenously produced antibody from degradation.^{77,78} It will be important to understand whether PrEP affects epithelial turnover and/or this recycling pathway, as new HIV-1 prophylactic mAbs in the pipeline use modifications to enhance FcRn recycling;⁸ therefore, the clearance and half-life of the new modifications may also be differentially modified by PrEP.”

We elected not to mention tight junctions in the possible mechanisms affecting antibody transport because we did not find any evidence of altered paracellular intestinal transport (as indicated by no inflammation or increases in LBP after PrEP uptake). Our data is more suggestive of alterations to intracellular transport pathways that mediate antibody recycling, and where TDF-FTC and its metabolites reside. To further study this, we are conducting a small pilot that collects rectal biopsies from PrEP users and non-PrEP users, which should allow us to examine by immunohistochemistry whether tight junctions remain intact. This approach was infeasible in the AMP samples, since only blood was collected.

Minor Comments:

1) When addressing processes such as the ones mediated by the cytochrome CYP450 (lines 77, lines 327-328), the authors should review their statement indicating that this is “drug transporter pathway”, rather these processes are metabolizing pathways or processes that result in drug or mAbs metabolism or degradation. Drug transport is the process which mediates permeability of drugs across biological membranes either by simple diffusion (membrane pore) and /or by a specific process involving a membrane-associated transporter.

Response: Thanks to the reviewer for pointing out our error. The CYP450 is not involved in TDF-FTC degradation so we have modified the sentence as follows:

“In addition, VRC01, like most mAbs, has no known processing by the enzymes that activate and process TDF-FTC.⁴⁵ TDF-FTC is not known to impair P glycoprotein, an efflux transporter that could mediate drug interactions”

2) Line 76 “ADA” should be spelled out. This acronym is spelled later on, line 211

Response: We now spell out anti-drug antibodies when it first appears.

Reviewer #2 (Remarks to the Author):

This is a well-written, important manuscript evaluating the effect of oral TDF/FTC PrEP on the PK of VRC01, an HIV-1 broadly neutralizing antibody that has been evaluated for preventive efficacy in the AMP study. Noteworthy results include the finding that VRC01 recipients who took PrEP had a significantly faster VRC01 clearance rate and lower VRC01 AUC compared to non-PrEP users – these effects are predicted to result in a 14% lower VRC01-mediated preventive efficacy among PrEP users. Additionally, the research team have done a thorough evaluation of potential mechanisms of these effects, and found that levels of I-FABP were elevated among PrEP users, suggesting increased epithelial intestinal permeability, and these elevated levels were positively associated with higher VRC01 clearance. These findings are highly significant, as this is the first study demonstrating an effect of oral TDF/FTC PrEP on the PK of a monoclonal antibody, and has important implications which are outlined in the discussion, including the need to assess PK of additional mAbs being tested for HIV prevention among PrEP users vs. non-users; the importance of assessing whether similar findings are seen in people living with HIV who are on antiretrovirals and participate in a treatment interruption trial using mAbs, and finally evaluating whether the PK of other prophylactic or therapeutic antibodies for other diseases in PrEP users is impacted.

The conclusions in this paper are well supported by the detailed methods and analyses presented in the manuscript. The authors point out that urine samples were not collected as part of AMP, and therefore they were not able to explore directly any tubular secretion or reabsorption detects of the mAb. Additionally, due to the large size the AMP study, they did not collect any mucosal samples to allow exploration of VRC01 clearance in intestinal secretions or mechanisms of intestinal pathology. In the discussion, the authors point to 2 possible mechanisms for how PrEP could affect VRC01 transport in the intestinal epithelium – affecting turnover of intestinal epithelial cells and/or impacting the endosomal transport and recycling of antibodies. It would strengthen the discussion if the authors could briefly state what additional studies are recommended to differentiate these two mechanisms in future studies.

Response: As described above, we have initiated a small pilot project that collects urine, rectal biopsies, rectal secretions, and blood from PrEP and non-PrEP users outside of AMP because the AMP study did not collect these specimens. We plan to use the blood to confirm that we can detect increases in IFABP in another cohort of PrEP users. Additionally, in mucosal biopsies we plan to examine several hypotheses, including 1) any changes in the epithelial invaginations and any shortening of the surface area, 2) epithelial cell death, and the distribution of goblet cells or absorptive epithelium, 3) the distribution of the FcRn, which transports antibody and albumin at the surface, and 4) any changes in antibodies or albumin in rectal secretions or urine of PrEP users, confirming increased permeability into the lumen or renal secretion issues. These experiments would help us understand some of the mechanisms for increased antibody clearance in PrEP users and confirm whether intestinal secretion accounts for the clearance. However, the details of these experiments are outside of the scope of this manuscript. To the discussion, we added:

“Studies of intestinal biopsies from PrEP users and non-PrEP users might help elucidate whether these or other mechanisms are at play”.

The methodology used in this study are well-described and in general are sufficient to be reproduced. Under the dried blood spot assay description, it would be helpful to note whether samples were frozen at sites and within what time window after sample collection. Also, it would be helpful to specify the lower limit of detection for TFV-DP in DBS here in the text.

Response: Thank you for the question. We now have added additional details under “DBS Assay” in the Supplementary Materials. Specifically,

“The methods described here are specific to testing for two PrEP drug anabolites, intraerythrocytic TFV-DP (tenofovir diphosphate) and FTC-TP (emtricitabine diphosphate).^{1,2}

TFV-DP is the active metabolite exists within cells after TDF is metabolized, and is responsible for inhibiting viral replication within infected cells; FTC-TP is the active intracellular form generated after FTC is metabolized within cells, and contributes to the overall inhibition of HIV replication via incorporation into viral DNA.

For drug concentration testing, 25 ul of blood from EDTA tubes is spotted five times onto 903 Protein Saver Cards (Whatman/GE Healthcare, Piscataway, NJ) (125 ul blood used in total). After spotting, the cards are dried for at least 2 hours and then placed in plastic bags and stored in a sample box with desiccant and humidity indicators. For short term storage, room temperature (<5 days) or 4C may be used; for longer term storage, -20C and -80C have been shown to have acceptable stability up to 5 years (reference 6.2, validation report).

For the extraction of analytes from DBS, a 3-mm diameter disk is punched from the blood spot on the card, using a micropuncher. A punch from a clean Protein Saver Card is performed in between each DBS sample in order to avoid analyte contamination from the previous DBS punch. The disk is placed in a microcentrifuge tube with 500 ul of 70:30 methanol–water solution. Following extraction, the lysed cellular matrix is subjected to solid phase extraction procedures to prepare the sample for liquid-chromatography tandem mass spectrometry. The validated quantifiable linear range for TFV-DP is 25–2000 fmol/sample and that for FTC-TP is 0.1–200 pmol/sample for a 3-mm punch. Because 400ul of the 500ul are assayed (comprising the “sample”), the lower limit of quantification of TFV-DP for a 3mm punch is 31.25 fmol/punch and that for FTC-TP is 0.125 pmol/punch. Stable labeled isotopic internal standards are used to ensure accuracy and precision in various cell matrices.

As previously reported, TFV-DP accumulates appreciably in red blood cells with repeated daily doses and has a long half-life of 17 days in this cellular compartment. Therefore, measurement of TFV-DP in DBS is useful to assess cumulative patterns of adherence over longer periods, and provides valuable insight into both dosing recency and cumulative doses from variable adherence patterns.^{3,4} In all subsequent analyses, we defined PrEP use status based on TFV-DP concentration in DBS because the decay kinetics and levels of effective use have been reasonably well established for TFV-DP.”

Here are some additional minor recommendations / comments for clarity:

Line 46 – AWARE trial showed that risk reduction counseling was not effective in reducing STIs - consider removing risk reduction counseling from the text

Response: We changed this to say “other risk reduction strategies.”

Line 76- define ADA

Response: We now spell out *anti-drug antibodies* (ADA) when it first appears.

Line 94 – would be helpful to specify the level of PrEP uptake and adherence in the AMP study here in the text.

Response: We added the percentage of uptake to the statement. We also added an estimate of adherence to the study population description:

“Among these PrEP users, at the last visit with confirmed PrEP usage, their DBS samples contained a median TFV-DP level of 1145 fmol/punch (range 540-2437), suggesting an average of at least 4 doses/week in the past 6-8 weeks.”²⁶”

Line 154 – change to creatinine clearance (not creatine clearance)

Response: We changed creatine to creatinine.

Line 226-227 The sentence “since creatinine clearance...” seems incomplete

Response: We corrected this statement. The sentence now reads:

“It is possible that PrEP-associated side effects may contribute to the observed differences in VRC01 PK between PrEP and non-PrEP users, since CrCl could be associated with mAb clearance, and the liver appears to be a major site for catalysis of Fc-containing antibodies,^{43,44} such as VRC01.”

Line 480 – Should “Samples were rerun in the CV of the duplicates was higher than 30% CV” be changed to “Samples were rerun if the CV of the duplicates was higher than 30% CV”?

Response: We corrected this via changing “in” to “if”.

Reviewer #3 (Remarks to the Author):

This study sought to understand why the clearance of the mAb VRC01 was faster in subjects on oral pre-exposure prophylaxis (PrEP) consisting of tenofovir and emtricitabine compared to subjects not on PrEP. There is no clear or obvious pharmacokinetic interaction, hence the novel science being conducted here is worthy of reporting. However, there are several items that need clarification or revision.

Specific Comments

Abstract

- third sentence, check units on “dose-normalized VRC01 serum concentrations”. 0.29 day/mL seems wrong. It should be something like 0.29 ug/mL/day, e.g.

Response: Units are correct. Added “area-under-the-curve” to help clarify.

- Just because the clearance was 14% faster and serum concentrations 14% lower doesn't always mean that efficacy will be 14% lower too. It depends on the receptor occupancy of VRC01 with its target receptor, the expression levels of that target receptor in patients with HIV-1, the disease burden of each individual, etc. Given that these are healthy subjects without HIV, could there be another physiological reason for inter-individual variability in the target receptor (CD4 binding site)? I understand the PT80 values were calculated to assess efficacy, but again, that metric incorporates the serum concentration.

Response: Thank you for bringing up this question. We agree with the reviewer that there could be mechanisms, other than neutralization, that contribute to the efficacy of VRC01. In addition, there could be inter-individual variability in serum concentrations and clearance of VRC01. We hence changed "VRC01-mediated prevention efficacy" to "VRC01 PT₈₀-mediated prevention efficacy" throughout the text and added "on average" to indicate that the effect was a population-level estimate.

Introduction

- More background information is needed about VRC01 itself and the mechanism by which it is believed to prevent HIV infection upon exposure. Simply stating that it is a CD4-binding site targeting mAb for prophylactic HIV prevention doesn't mean much to the lay reader. A brief 2-3 sentence background on why inhibiting (I'm assuming VRC01 binds CD4 and inhibits this site from interacting with some other ligand) is believed (with citations) to prevent HIV infection and/or transmission.

Response: We added more background about VRC01 to the introduction.

Specifically, in the second paragraph, we state:

"VRC01 was originally discovered in a person living with HIV-1 for more than 15 years, who maintained viral control without use of antiretroviral therapy (ART).^{4,7} VRC01 binds the HIV envelope site that interacts with the CD4 molecule on target cells, has the capacity to neutralize a broad range of HIV-1 strains in vitro and has demonstrated protection in multiple non-human primate challenge studies."

- Second paragraph...the dosing schedule of every 8 weeks, how was that optimized, but the dose amount (10 mg/kg or 30 mg/kg) not? Please cite the study(s) that selected 8 weeks as the optimal dosing interval. This prolonged schedule is atypical of most therapeutic mAbs and in particular with VRC01 having a known half-life of 15 days.

Response: We added that the dosing and schedule were based on a phase 1 studies to the second paragraph of the introduction, stating:

"VRC01 every 8 weeks at a dose of either 10 or 30 mg/kg or placebo, for 10 infusions in total, with dose and schedule determined based on previous early phase clinical studies."^{11,12}

- Second paragraph, you state "...viruses that were sensitive to VRC01." Is it known if certain strains of HIV are more or less sensitive to VRC01, presumably based on a yet-to-be-stated mechanism of action?

Response: Thanks for the question. Yes, the VRC01 neutralization sensitivity of a given HIV strain can be evaluated in vitro, quantified by the needed concentration of the clinical lot of VRC01 to neutralize the pseudotyped HIV strain using the TZM-bl neutralization assay. This in vitro neutralization sensitivity is often referred to as the 80% Inhibitory Concentration (IC80) as described in the paper. In addition, excellent research has been done to understand HIV-1 Env

genotypic/amino acid features that affect VRC01 resistance, including but are limited to the work listed below. Most recently, Magaret et al. (2019) investigated features predicting neutralization sensitivity or resistance, including 26 surface-accessible residues in the VRC01 and CD4 binding footprints, the length of gp120, the length of Env, the number of cysteines in gp120, the number of cysteines in Env, and 4 potential N-linked glycosylation sites. Many of the residues identified as highly predictive and are supported by experimental evidence as being important for VRC01 binding. Four of the top-ranked AAs found in this study (D279, N280, R456, and G459) have been shown to be sites of common interactions with potent VRC01-like Abs, and D279 and E459 have been identified as making critical interactions with VRC01. Moreover, mutation of residue D279 to E279 (D279E) was shown to be part of the VRC01 escape pathway within the donor from whom VRC01 was isolated.

We are providing this in response to our reviewer's question in this letter, but not in the manuscript because neutralization is not the focus of our presented work.

- Zhou T, Georgiev I, Wu X, Yang ZY, Dai K, Finzi A, et al. Structural basis for broad and potent neutralization of HIV-1 by antibody VRC01. *Science*. 2010;329(5993):811–7. 10.1126/science.1192819
- Li Y, O'Dell S, Walker LM, Wu X, Guenaga J, Feng Y, et al. Mechanism of neutralization by the broadly neutralizing HIV-1 monoclonal antibody VRC01. *J Virol*. 2011;85(17):8954–67. 10.1128/JVI.00754-11
- Lynch RM, Wong P, Tran L, O'Dell S, Nason MC, Li Y, et al. HIV-1 fitness cost associated with escape from the VRC01 class of CD4 binding site neutralizing antibodies. *J Virol*. 2015;89(8):4201–13.
- Guo D, Shi X, Arledge KC, Song D, Jiang L, Fu L, et al. A single residue within the V5 region of HIV-1 envelope facilitates viral escape from the broadly neutralizing monoclonal antibody VRC01. *J Biol Chem*. 2012;287(51):43170–9. 10.1074/jbc.M112.399402
- Wibmer CK, Bhiman JN, Gray ES, Tumba N, Abdool Karim SS, Williamson C, et al. Viral escape from HIV-1 neutralizing antibodies drives increased plasma neutralization breadth through sequential recognition of multiple epitopes and immunotypes. *PLoS Pathog*. 2013;9(10):e1003738
- Utachee P, Isarangkura-na-ayuthaya P, Tokunaga K, Ikuta K, Takeda N, Kameoka M. Impact of amino acid substitutions in the V2 and C2 regions of human immunodeficiency virus type 1 CRF01_AE envelope glycoprotein gp120 on viral neutralization susceptibility to broadly neutralizing antibodies specific for the CD4 binding site. *Retrovirology*. 2014;11:32 10.1186/1742-4690-11-32
- Magaret CA, Benkeser DC, Williamson BD, Borate BR, Carpp LN, Georgiev IS, Setliff I, Dingens AS, Simon N, Carone M, Simpkins C, Montefiori D, Alter G, Yu WH, Juraska M, Edlefsen PT, Karuna S, Mgodini NM, Edugupanti S, Gilbert PB. Prediction of VRC01 neutralization sensitivity by HIV-1 gp160 sequence features. *PLoS Comput Biol*. 2019 Apr 1;15(4):e1006952. doi: 10.1371/journal.pcbi.1006952.

Results

- “Markers of Hepatic and Renal...”, third paragraph, you state faster clearance of VRC01 were not “temporarily” linked. Do you mean “temporally”?

Response: Yes, we did mean temporally and corrected it. Thank you for noticing this error.

- “Impact of intestinal permeability on VRC01 PK”, second sentence: “...at this time-point, only among PrEP users (not PrEP users).”. did you mean to say “..(not non-PrEP users)”?

Response: Yes, we did mean “not non-PrEP users” and corrected it. Thank you for noticing this error.

- Was the impact of serum albumin concentrations assessed for correlation with VRC01 clearance? There is plenty of literature on the correlation of albumin and mAb clearance due to increased general protein turnover/catabolism.

Response: Thank you for the important question. As the reviewer indicates, we are particularly interested in the assessment of albumin, as it will help us address issues of potential renal tubular secretion, intestinal luminal secretion, and assess transport of the FcRn, which recycles both albumin and antibodies. We are eager to monitor albumin in blood and the additional compartments in the pilot cohort of PrEP users and non-PrEP users that we described earlier in our responses. Because only serum is available for AMP samples, at the reviewer’s request we assessed the serum albumin concentrations in PrEP users and non-PrEP users. However, interpreting serum albumin levels is difficult, as they combine multiple pathways (liver synthesis, renal absorption, intestinal recovery, skin and muscular degradation). We found that serum albumin was comparable at baseline, the “early visit on PrEP”, and the “last visit on PrEP” for PrEP users (left panel, Figure below). In addition, we did not find any significant correlation between serum albumin and VRC01 CI or AUC. These results confirmed the lack of evidence for kidney or liver damage, which are primary mechanisms for lowering albumin levels.

Unfortunately, the albumin levels in these participants are rather variable, and we are likely not powered to detect 10% differences, which would be expected from the contributions of intestinal permeability to systemic albumin levels. The results so far supported the conclusion that the gastrointestinal permeability associated with PrEP is selective, not mediating increases in LBP or decreases in serum albumin concentrations.

We did not include this analysis in our manuscript for the following reasons: 1) the most useful analysis of albumin would be in the context of urine and rectal secretions, to explore tubular secretion and any gastrointestinal leakage; 2) as mentioned in our earlier response, we plan to examine albumin in our future studies of urine and rectal secretions collected from PrEP users outside of the AMP study; 3) monoclonal antibodies can bind to albumin in a non-specific manner, which could make it challenging to interpret the relationship between albumin concentrations and antibody clearance accurately; and, 4) in our experiments, the measurements of albumin

concentrations seemed rather variable, and we did not understand the exact cause of the high albumin levels in non-PrEP users at baseline. This could be suggestive of albumin aggregation in the freeze thawed samples, dehydration of the participants, or issues with the error in the 1:2,000,000 dilution required for the assay, which is optimized to detect small albumin concentrations in urine. We needed more time to feel confident in these albumin results before publication.

Methods

- More information is needed about the LC-MS/MS assays used to measure TDF, TFV and emtricitabine (FTC), and the metabolites TFV-dp and FTC-TP. Was a single LC-MS/MS assay used to measure all analytes, or because two DBS punches were used, were there two separate assays? More information is needed about this...

Response: Thank you for the question. We have now added more details about the DBS assay and streamlined the description in the supplementary materials:

“The methods described here are specific to testing for two PrEP drug metabolites, intraerythrocytic TFV-DP (tenofovir diphosphate) and FTC-TP (emtricitabine diphosphate).^{1,2}

TFV-DP is the active metabolite exists within cells after TDF is metabolized, and is responsible for inhibiting viral replication within infected cells; FTC-TP is the active intracellular form generated after FTC is metabolized within cells, and contributes to the overall inhibition of HIV replication via incorporation into viral DNA.

For drug concentration testing, 25 ul of blood from EDTA tubes is spotted five times onto 903 Protein Saver Cards (Whatman/GE Healthcare, Piscataway, NJ) (125 ul blood used in total). After spotting, the cards are dried for at least 2 hours and then placed in plastic bags and stored in a sample box with desiccant and humidity indicators. For short term storage, room temperature (<5 days) or 4C may be used; for longer term storage, -20C and -80C have been shown to have acceptable stability up to 5 years (reference 6.2, validation report).

For the extraction of analytes from DBS, a 3-mm diameter disk is punched from the blood spot on the card, using a micropuncher. A punch from a clean Protein Saver Card is performed in between each DBS sample in order to avoid analyte contamination from the previous DBS punch. The disk is placed in a microcentrifuge tube with 500 ul of 70:30 methanol-water solution. Following extraction, the lysed cellular matrix is subjected to solid phase extraction procedures to prepare the sample for liquid-chromatography tandem mass spectrometry. The validated quantifiable linear range for TFV-DP is 25–2000 fmol/sample and that for FTC-TP is 0.1–200 pmol/sample for a 3-mm punch. Because 400ul of the 500ul are assayed (comprising the “sample”), the lower limit of quantification of TFV-DP for a 3mm punch is 31.25 fmol/punch and that for FTC-TP is 0.125 pmol/punch. Stable labeled isotopic internal standards are used to ensure accuracy and precision in various cell matrices.

As previously reported, TFV-DP accumulates appreciably in red blood cells with repeated daily doses and has a long half-life of 17 days in this cellular compartment. Therefore, measurement of TFV-DP in DBS is useful to assess cumulative patterns of adherence over longer periods, and provides valuable insight into both dosing recency and cumulative doses from variable adherence patterns.^{3,4} In all subsequent analyses, we defined PrEP use status based on TFV-DP concentration in DBS because the decay kinetics and levels of effective use have been reasonably well established for TFV-DP.”

Table 2: Why does AUC have units of “Days/mL”?

Response: This is because the AUC is dose-normalized as indicated in the text and figures. Specifically, the area under the curve is calculated as the dose-normalized concentration (1/mL) multiplied by the time-interval (days) of the concentration-time curve, where concentration before dose-normalization is in the unit of mg/mL and the dose amount is in the unit of mg.

Table 2/Figure 3: V_p in Table 2, using TMLE-predicted, covariate-adjusted mean values, had a nonsignificant difference between PrEP vs non-PrEP users (5.24 L vs 4.88 L). However, in Figure 3b, the visualization of this is drastic and presents as a clearly distinct profile.

Response: Thank you for bringing up this observation. We agree with our reviewer that the visualization of the covariate-adjusted mean values of V_p looked different between PrEP and non-PrEP users. The reason why the statistical test of this difference was not significant was because the precision of the comparison was dependent not only on the inter-individual variability, which seemed small based on the visualization, but also the estimation error of each of the covariate-adjusted values of V_p , which is not shown in the visualization but was accounted for in the testing. Due to the sparse sampling of the PK time-points and the relatively small sample size, the latter is rather large for V_p and hence the low efficiency in the comparison. We also performed additional sensitivity analysis that removed one outlier with particularly high estimation error in V_p ; however, as reported in the manuscript:

“Consistent results were observed in a sensitivity analysis that excluded data from one PrEP user (ID=21) with unstable PK parameter estimates (Supplementary Table 4).”

Reviewers' Comments:

Reviewer #1:

Remarks to the Author:

The present reviewer is satisfied with the response to the comments which have been thoroughly addressed by the authors.

Reviewer #2:

Remarks to the Author:

I have reviewed the revisions and my comments have been adequately addressed. Thank you.